# Orgo-Seq integrates single-cell and bulk transcriptomic data to identify cell type specific-driver genes associated with autism spectrum disorder

Elaine T. Lim [1,2,3,4 ✉], Yingleong Chan[1,2,3], Pepper Dawes[1,2,3,4], Xiaoge Guo[5,6], Serkan Erdin [7,8,9], Derek J. C. Tai[7,8,9,10], Songlei Liu [5,6], Julia M. Reichert[1,2,3], Mannix J. Burns [1,2,3,4], Ying Kai Chan[5,6], Jessica J. Chiang[5,6], Katharina Meyer[5], Xiaochang Zhang [11,12], Christopher A. Walsh [9,13,14,15,16,17], Bruce A. Yankner[5], Soumya Raychaudhuri [9,18,19,20], Joel N. Hirschhorn[5,9,21,22], James F. Gusella[5,8,9,23], Michael E. Talkowski [7,8,9,10,17,24] & George M. Church [5,6 ✉]

Cerebral organoids can be used to gain insights into cell type specific processes perturbed by genetic variants associated with neuropsychiatric disorders. However, robust and scalable phenotyping of organoids remains challenging. Here, we perform RNA sequencing on 71 samples comprising 1,420 cerebral organoids from 25 donors, and describe a framework (Orgo-Seq) to integrate bulk RNA and single-cell RNA sequence data. We apply Orgo-Seq to 16p11.2 deletions and 15q11–13 duplications, two loci associated with autism spectrum disorder, to identify immature neurons and intermediate progenitor cells as critical cell types for 16p11.2 deletions. We further applied Orgo-Seq to identify cell type-specific driver genes. Our work presents a quantitative phenotyping framework to integrate multi-transcriptomic datasets for the identification of cell types and cell type-specific co-expressed driver genes associated with neuropsychiatric disorders.

[1] Program in Bioinformatics and Integrative Biology, University of Massachusetts Chan Medical School, Worcester, MA 01605, USA. [2] Department of Neurology, University of Massachusetts Chan Medical School, Worcester, MA 01605, USA. [3] NeuroNexus Institute, University of Massachusetts Chan Medical School, Worcester, MA 01605, USA. [4] Department of Molecular, Cell and Cancer Biology, University of Massachusetts Chan Medical School, Worcester, MA 01605, USA. [5] Department of Genetics, Blavatnik Institute, Harvard Medical School, Boston, MA 02115, USA. [6] Wyss Institute for Biologically Inspired Engineerin, Harvard University, Boston, MA 02115, USA. [7] Psychiatric and Neurodevelopmental Genetics Unit, Center for Genomic Medicine, Massachusetts General Hospital, Boston, MA 02114, USA. [8] Molecular Neurogenetics Unit, Center for Genomic Medicine, Massachusetts General Hospital, Boston, MA 02114, USA. [9] Program in Medical and Population Genetics, Broad Institute of MIT and Harvard, Cambridge, MA 02115, USA. [10] Department of Neurology, Massachusetts General Hospital, Boston, MA 02114, USA. [11] Department of Human Genetics, The University of Chicago, Chicago, IL 60637, USA. [12] The Grossman Neuroscience Institute, The University of Chicago, Chicago, IL 60637, USA. [13] Division of Genetics and Genomics, Boston Children's Hospital, Boston, MA 02115, USA. [14] Manton Center for Orphan Disease Research, Boston Children's Hospital, Boston, MA 02115, USA. [15] Howard Hughes Medical Institute, Boston, MA 02115, USA. [16] Department of Pediatrics, Harvard Medical School, Boston, MA 02115, USA. [17] Department of Neurology, Harvard Medical School, Boston, MA 02115, USA. [18] Center for Data Sciences, Brigham and Women's Hospital and Harvard Medical School, Boston, MA 02115, USA. [19] Division of Rheumatology and Genetics, Brigham and Women's Hospital and Harvard Medical School, Boston, MA 02115, USA. [20] Centre for Genetics and Genomics Versus Arthritis, Manchester Academic Health Science Centre, University of Manchester, Manchester M13 9PL, UK. [21] Division of Endocrinology, Boston Children's Hospital, Boston, MA 02115, USA. [22] Center for Basic and Translational Obesity Research, Boston Children's Hospital, Boston, MA 02115, USA. [23] Harvard Stem Cell Institute, Harvard University, Cambridge, MA 02138, USA. [24] Stanley Center for Psychiatric Research, Broad Institute of MIT and Harvard, Cambridge, MA 02115, USA. ✉email: elaine.lim@umassmed.edu; gchurch@genetics.med.harvard.edu

Recent advances in cerebral organoid models differentiated from human induced pluripotent stem cells (iPSCs) demonstrated that these in-vitro systems comprise of many cell types found in the developing human fetal brain[1–4], and show great promise as a system for identifying cell types and cell-type-specific molecular processes that are perturbed in neurodevelopmental and neuropsychiatric disorders such as microcephaly and autism spectrum disorders (ASD)[2,5,6]. Identifying the cell types and cell type-specific co-expressed genes that are perturbed in disease-associated loci allows us to perform direct experiments on relevant cell types to understand molecular processes that are important in disease.

There are key challenges to the application of cerebral organoids for identifying cell types and cell type-specific processes that are perturbed in complex neuropsychiatric disorders. Prior literature has demonstrated that the cerebral organoids are comprised of many different cell types found in the human brain, and individual organoids can be heterogeneous in their cell type compositions detected using single-cell RNA sequencing (scRNA-seq)[1]. This poses additional challenges for detecting robust cellular and molecular differences between cerebral organoids differentiated from individuals with different genetic backgrounds. To address this key challenge, we differentiated a large number of 1420 organoids from 25 individuals with diverse backgrounds (71 samples with 20 organoids per sample), to systematically quantify and identify the inherent variability in whole-transcriptome bulk RNA sequence (bRNA-seq) data from the organoids.

Another challenge is the robust detection of cell type-specific co-expressed genes that are perturbed in donor-derived cerebral organoids. One approach is to use scRNA-seq to perform unbiased discovery of critical cell types and cell type-specific co-expressed genes associated with diseases[1,7–14]. However, current scRNA-seq technologies capture only 10–20% of all transcripts[15], and cell type-specific co-expression of many disease-associated genes might not be detectable with scRNA-seq. For instance, within the 16p11.2 locus associated with ASD, the expression for only 2 of the 29 genes in the locus (*QPRT* [NCBI Gene ID: 23475] and *ALDOA* [NCBI Gene ID: 226]) were detected among the 10 major cell type clusters identified using scRNA-seq on cerebral organoids[1].

Here we developed a quantitative phenotyping framework (termed Orgo-Seq, Fig. 1), which allows researchers to identify cell type-specific co-expressed driver genes by integrating bRNA-seq data from donor-derived organoids with large-scale scRNA-seq data from brain organoids and fetal brains. This allows us to overcome the limitations with current scRNA-seq technologies, and at the same time, leverage on the strengths of large-scale scRNA-seq datasets that have been previously generated or will be generated in the future for unbiased discoveries of cell types and cell type-specific co-expressed driver genes.

We applied Orgo-Seq for two ASD-associated copy number variants (CNVs) in the 16p11.2 and 15q11–13 loci[16–18], by integrating three sets of transcriptomics datasets: bRNA-seq data that we generated from donor-derived cerebral organoids, previously published scRNA-seq data from cerebral organoids and fetal brains[1,13,19,20], and previously published bRNA-seq data from human post-mortem brain samples in the BrainSpan Project[21]. Using an initial scRNA-seq dataset from 66,889 single cells[1], we initially observed that neuroepithelial cells are perturbed in donor-derived cerebral organoids from individuals with deletions in 16p11.2 compared to individuals without the deletions, and that three of the genes in the locus (*YPEL3* [NCBI Gene ID: 83719], *KCTD13* [NCBI Gene ID: 253980] and *INO80E* [NCBI Gene ID: 283899]) are likely to be cell type-specific candidate driver genes functioning in neuroepithelial cells.

Using a larger scRNA-seq dataset comprising of 190,022 cells[19] from brain organoids differentiated using eight different protocols and fetal brains[1,7–13,22], and two neurodevelopmental maps constructed from scRNA-seq on brain organoids and fetal brains to fine-map the critical cell types[13,20], we replicated the critical cell type that was initially discovered, and were able to pinpoint the identity of the critical cell type more precisely during neurodevelopment to immature neurons and intermediate progenitor cells for the 16p11.2 locus. We also replicated our initial results that *YPEL3*, *KCTD13,* and *INO80E* are cell type-specific co-expressed driver genes for the 16p11.2 locus. Our work presents a quantitative framework to identify cell types and cell type-specific driver genes in a complex disease by integrating bRNA-seq and scRNA-seq from donor-derived cerebral organoids and human post-mortem brains using Orgo-Seq.

## Results

**Low variability in bRNA-seq data from pooling individual cerebral organoids**. It has been previously reported that one key challenge impeding the use of cerebral organoids as a system is the high variability when comparing single cells from the organoids or single organoids from a few donors[1]. To address these issues, we obtained iPSCs and differentiated 1420 cerebral organoids from 25 individuals: 12 control donors (termed "controls") and 13 donors with 16p11.2 deletions or 15q11–13 duplications (termed "cases"), shown in Table 1. DNA was extracted from the iPSCs and CNV detection was performed on iPSCs from all donors using array comparative genomic hybridization or aCGH; whole-exome sequencing to detect smaller exonic CNVs; and whole-genome sequencing to detect the breakpoints of the CNVs (Supplementary Data 1–4). All controls were confirmed not to harbor any CNVs within the two ASD-associated loci in 16p11.2 and 15q11–13.

We differentiated cerebral organoids using the 25 iPSCs for 46 days, by adapting a previously described method[23] (Table 1, Supplementary Fig. 1), and performed RNA sequencing on 1–3 replicates for each donor, resulting in a total of 71 samples (Fig. 2, Supplementary Data 1). We compared the standard deviations in gene expression between replicates for each individual (intra-individual), as well as across organoids differentiated from different individuals (inter-individual). We found that there were 860 genes (7.6% of all expressed genes) that showed high intra-individual variability, and 869 genes (7.7% of all expressed genes) that showed high inter-individual variability (Supplementary Figs. 2 and 3A, B). These genes with high intra-individual or inter-individual variability were enriched in processes involved in nervous system development, neurogenesis, and cell differentiation (Supplementary Data 5), which might contribute to the inherent variability in spontaneous differentiation of these cerebral organoids. These highly variable genes were not enriched for genes with genetic or genomic associations with ASD[24,25]. For our downstream analyses, we removed these highly variable genes and focused on a smaller, robust group of genes with low technical variability in expression, and there are 9978 such unique genes that were detected in the organoids.

We found that there were low variability and high mean intra-individual correlations $r^2$ of 0.97 and mean inter-individual correlations $r^2$ of 0.94 in bRNA-seq data generated from the cerebral organoids using our approach (Supplementary Fig. 3C–H). Similar to previous reports[5,26], we observed significantly higher intra-individual correlations compared to inter-individual correlations (Wilcoxon $P = 1.03 \times 10^{-7}$), confirming that bRNA-seq data from the cerebral organoids can reflect biological differences between individuals that are not due to technical differences between replicates from the same individual.

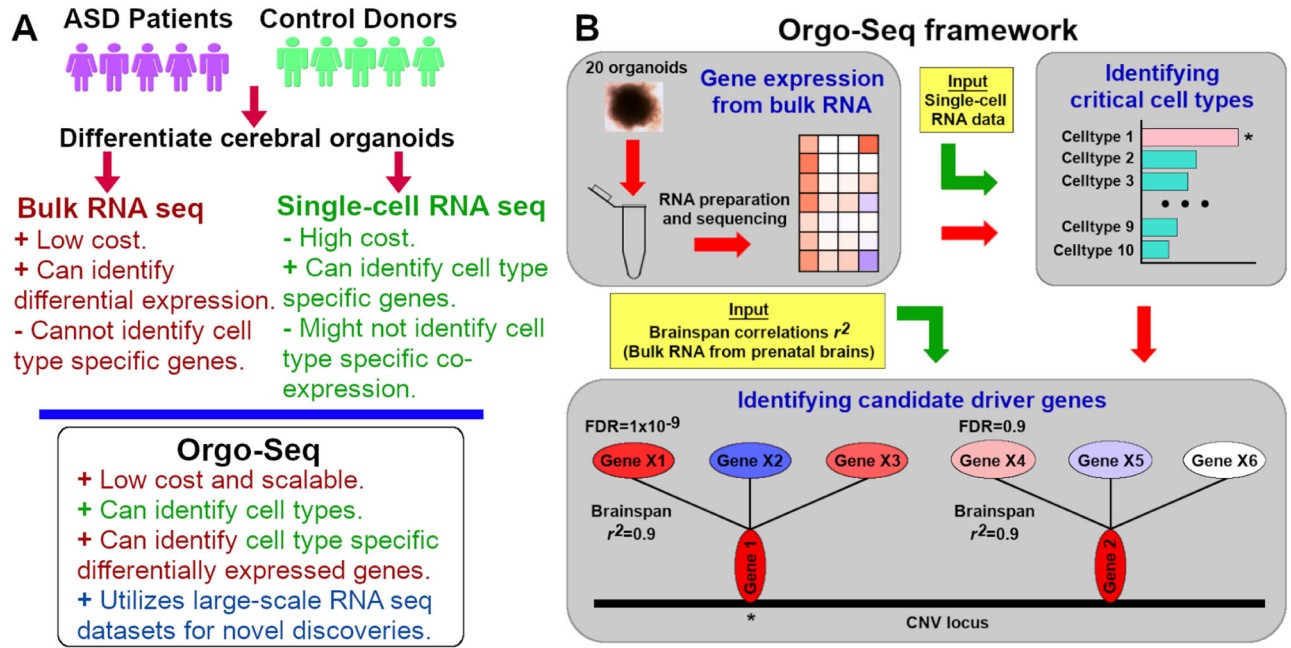

**Fig. 1 Orgo-Seq framework to identify cell type-specific co-expressed driver genes. A** Figure illustrating the strengths and weaknesses of bRNA-seq and scRNA-seq, and what Orgo-Seq can achieve by integrating both types of datasets. **B** A schematic of the Orgo-Seq framework to integrate bRNA-seq data from patient-derived brain organoids with scRNA-seq data from control brain organoids, for the discovery of critical cell types and cell type-specific driver genes.

**Table 1 Details of the iPSC lines used in our study.**

| Details of iPSC lines | Number of iPSC lines |
|---|---|
| Source/Biorepository | Personal Genome Project (1), Coriell (3), ATCC (7), RUDCR (9), Harvard Stem Cell Institute (5) |
| Ethnicity of donors | White (18), Black (3), Asian (2), Hispanic (2) |
| Biological sex of donors | Male (13), Female (12) |
| Diagnosis of ASD | Yes (7), No (18) |
| Tissue of origin | Fibroblast (12), Peripheral Vein (1), Bone Marrow (7), Peripheral Blood Mononuclear Cells (5) |
| Type of reprogramming | Sendai (13), Episomal (12) |
| CNVs | None (12), 16p11.2 deletions (9), 15q11–13 duplications (4) |

The table shows the details and numbers of the iPSC lines (1 clone from each line) in our study.

We used variancePartition[27] and principal components analyses to identify the sources of variation in the RNA sequence data from the organoids (Supplementary Fig. 4A, B), and found that most of the variation in gene expression (88%) could be accounted for by the first principal component (PC1) alone (Supplementary Fig. 5A–G). We further observed that age, the origin of the sample, and the type of reprogramming are significantly correlated with PC1 alone, but not with the second or third principal component (Supplementary Fig. 6).

**Transcriptome data in cerebral organoids accurately reflect copy number changes.** It was previously reported that bRNA-seq data from the cerebral organoids are highly correlated with bRNA-seq data from fetal brains[5], and we similarly observed high correlations between the bRNA-seq data from the cerebral organoids and fetal brains from the BrainSpan Project (Supplementary Fig. 7). In the absence of fetal brains with 16p11.2 deletions, we can effectively use cerebral organoids as a model system for identifying mutation-specific transcriptomic processes that are important in human neurodevelopmental diseases. The 16p11.2 locus encompasses 29 genes, and 22 of these genes are expressed in the organoids. In our study, there are three individuals with ASD and 16p11.2 deletions

(whom we termed as "probands"), six individuals with 16p11.2 deletions but were not clinically diagnosed with ASD (whom we termed as "resilient" individuals), and 12 control unaffected individuals without 16p11.2 deletions (Table 1). We further checked the first two principal components, but did not observe major stratification between the cases and controls (Supplementary Fig. 8).

We performed three sets of differential expression analyses on RNA sequence data from cerebral organoids differentiated from these individuals. SetA comparing all nine individuals with 16p11.2 deletions with 12 control individuals without 16p11.2 deletions; SetP comparing the three probands with ASD and 16p11.2 deletions with 12 control individuals without 16p11.2 deletions; and SetD analyses comparing only the individuals with 16p11.2 deletions: three probands with 16p11.2 deletions versus six resilient individuals with 16p11.2 deletions. We observed 2681 genes with FDR ≤ 0.05 in the SetA comparison, and 1853 genes with FDR ≤ 0.05 in the SetP comparison.

If RNA sequence data from the cerebral organoids can accurately reflect the underlying genetic mutations in the DNA (hemizygous deletions in the 16p11.2 locus or duplications in the 15q11–13 locus), then we should be able to reproduce the observation in peripheral tissue and mouse cortex that many of the genes in the 16p11.2 locus are down-regulated with fold

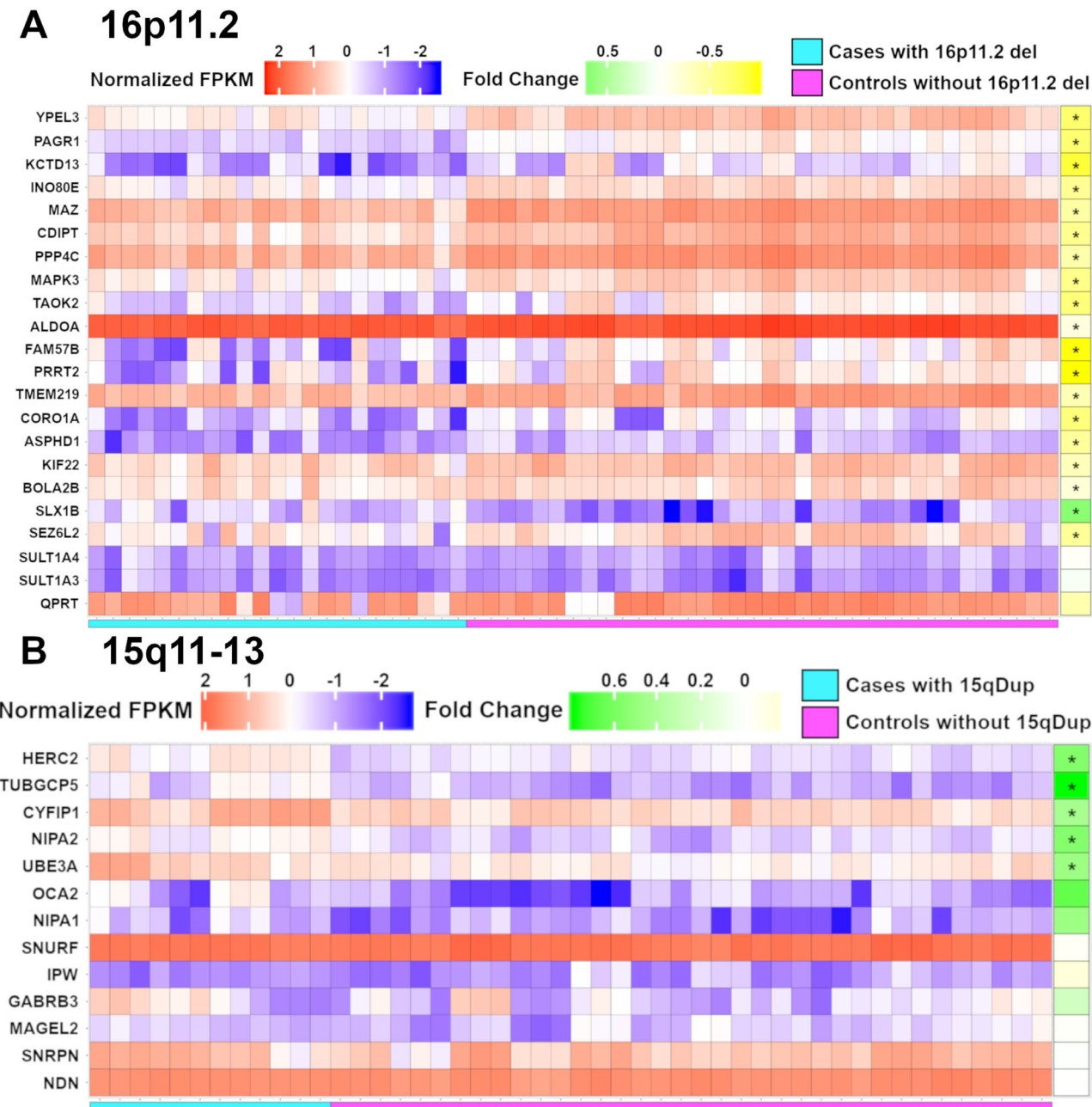

**Fig. 2 Expression of the gene products in the 16p11.2 and 15q11-13 loci.** All data were analyzed from 71 bRNA-seq samples over 25 donors. **A** Heatmap representation of the normalized expression (FPKM) for all samples across the 22 genes in the 16p11.2 locus. The fold change is represented as a green-yellow heatmap. An asterisk on the "Fold Change" heatmap indicates significant differential expression of the gene with FDR ≤ 0.05. **B** Heatmap representation of the normalized expression (FPKM) for all samples across the 13 genes in the 15q11–13 locus. The fold change is represented as a green-yellow heatmap. An asterisk on the "Fold Change" heatmap indicates significant differential expression of the gene with FDR ≤ 0.05.

changes of ~0.5 in the cases compared to controls[28]. For the SetA comparison, 19 of the 22 genes in the 16p11.2 locus (excluding *SULT1A4* [NCBI Gene ID: 445329], *SULT1A3* [NCBI Gene ID: 6818], and *QPRT* [NCBI Gene ID: 23475]) are significantly differentially expressed with FDR ≤ 0.05. The average fold-change for the 19 significantly differentially expressed genes in the 16p11.2 locus in the SetA comparison is 0.73. Seventeen of these 19 genes are also significantly differentially expressed in the smaller SetP comparison, with an average fold-change of 0.64. We did not detect a second genetic factor outside the 16p11.2 locus that contributes to increased risk for ASD, in addition to the 16p11.2 deletion background. Larger numbers of individuals with 16p11.2 deletions (with or without clinical ASD diagnoses) will be needed to identify

a second genetic hit with small effects, or it might be possible that the second hit is driven by non-genetic factors or by genes that are not expressed in cerebral organoids. However, we can exclude the hypothesis that there is a second genetic hit with large effects given our current sample sizes (Supplementary Data 6).

Out of the 25 individuals in our study, there are four individuals with ASD and 15q11–13 duplications, and 12 control unaffected individuals without 15q11–13 duplications (Table 1), and we similarly performed whole-transcriptome RNA sequencing on cerebral organoids differentiated from these individuals in triplicates (Supplementary Data 1). There are 16 genes that are significantly differentially expressed in the individuals with ASD and 15q11–13 duplications versus unaffected control individuals

with FDR ≤ 0.05. Out of the 16 genes, five of them are found in the 15q11–13 locus (*HERC2* [NCBI Gene ID: 8924], *TUBGCP5* [NCBI Gene ID: 114791], *CYFIP1* [NCBI Gene ID: 23191], *NIPA2* [NCBI Gene ID: 81614] and *UBE3A* [NCBI Gene ID: 7337]). The average fold-change for the 5 genes in the 15q11–13 locus that are significantly differentially expressed is 1.48, which closely reflects the 1.5-fold change in copy number across the locus, suggesting that the RNA sequence measurements are robust and quantitative. We did not detect smaller duplications that might encompass only a subset of these genes in the 15q11–13 locus for these individuals with ASD, using aCGH and whole-exome sequencing (Supplementary Data 2, 3).

**Data integration of bRNA-seq data from donor-derived cerebral organoids and scRNA-seq data from control organoids identifies critical cell types for 16p11.2 deletions and 15q11–13 duplications.** Deletions in 16p11.2 are significantly associated with ASD but not with schizophrenia, whereas duplications in 16p11.2 are associated with both ASD and schizophrenia[6,29,30]. Clinical studies have shown that individuals with 16p11.2 deletions have increased brain sizes, and individuals with duplications in the same locus have decreased brain sizes[29,31,32]. Mouse models with 16p11.2 deletions or duplications similarly show an increase or reduction in brain sizes and in the proportions of neural progenitor cells[33–35]. A systematic perturbation of all genes in the 16p11.2 locus using head sizes as the phenotypic readout in zebrafish identified *KCTD13* as the only driver gene in the locus modulating the proportion of neural progenitor cells[36]. However, recent studies in mice and zebrafish with deleted *KCTD13* did not observe increased brain sizes or neurogenesis in these mutant animal models[37,38]. In the absence of human fetal brains with 16p11.2 deletions that could be used to resolve these conflicting results from animal models[39], the use of donor-derived cerebral organoids could be good models to provide supporting results.

To accomplish this, we would have to identify which cell-type-specific co-expressed gene(s) from the donor-derived cerebral organoids are misregulating the proportions of critical cell types in cases versus controls (Fig. 1B). We developed a two-step solution where we first identified the critical cell types that were disproportionately affected in cases versus controls using bRNA-seq data from the donor-derived cerebral organoids, and a second step where we identified which of the genes in the CNV loci were disproportionately misregulating cell type-specific expression of genes outside the CNV loci between cases versus controls.

There are two general approaches to identifying critical cell types from bRNA-seq data: deconvolution methods such as CIBERSORT and CIBERSORTx, or cell type enrichment methods such as xCell[40,41]. Previously, when using bRNA-seq data from pure cell types as a reference panel, it was shown that a cell type enrichment approach (xCell) outperforms a deconvolution approach (CIBERSORT)[40]. We sought to develop a cell type enrichment-based approach for bRNA-seq data from cerebral organoids, by using scRNA-seq data from brain organoids and fetal brains as a reference panel. We developed a statistic termed CellScore, which is the difference between the weighted sum of all cell-type-specific genes and the weighted sum of all non-cell type-specific genes for each cluster of cell types, and the weights are the $-\log_{10}(P\text{-values})$ from our differential expression results in cerebral organoids. This allows us to identify transcriptomic signatures arising from the cell type-specific genes for each cluster, rather than the non-cell type-specific genes contributing to multiple clusters. We evaluated the significance of our observed CellScores using permutations (Supplementary Fig. 9).

We obtained scRNA-seq data from a publication by Quadrato et al. that found 10 major cell type clusters (c1–10) in 3-month-old and 6-month-old cerebral organoids from a control individual[1]. We separated the lists of genes for each cell-type clusters into a set of cell type-specific genes (ranging from 47 to 266 genes; Supplementary Data 7) that uniquely identifies each cluster of cell types, and a set of non-cell type-specific genes that are found in multiple clusters (ranging from 12 to 49 genes; Supplementary Data 7). When we calculated CellScores from the 16p11.2 SetA comparison, we found that the cluster comprising of neuroepithelial cells (c9) and unknown cluster (c6) were significantly perturbed ($P(\text{CellScore}) = 1.4 \times 10^{-3}$ and $P(\text{CellScore}) < 1 \times 10^{-6}$ respectively, Fig. 3A, Supplementary Data 8).

A recent study by Tanaka et al. had re-analyzed 190,022 cells from brain organoids differentiated using eight different protocols and fetal brains[1,7–13,22], and identified 24 cell type clusters[19]. We systematically compared the percentage overlaps among genes across the 24 cell type clusters to identify 11 unique clusters (CC1–11; Supplementary Data 9). We calculated CellScores using the 11 cell type clusters for 16p11.2, and found that there was only the cell type cluster comprising of cortical excitatory neurons (CC3) that had an FWER ≤ 0.05 ($P(\text{CellScore}) < 1 \times 10^{-6}$, Fig. 3C). Interestingly, we did not observe any association for 16p11.2 with the neuroepithelial cell cluster (CC7) in the Tanaka study[19] ($P(\text{CellScore}) = 0.04$).

When we calculated CellScores from the 15q11–13 data using the Quadrato and Tanaka datasets[1,19], we found that there were no clusters that were significantly perturbed with FWER ≤ 0.1 (Fig. 3C, D, Supplementary Data 8).

**Comparison to isogenic 16p11.2-derived 2-dimensional models show that 16p11.2 donor-derived cerebral organoids recapitulate signatures in neural stem cells more closely than induced neurons.** We previously engineered reciprocal deletion and duplication of 16p11.2 in an isogenic human iPSC line by targeting the flanking segmental duplications with CRISPR/Cas9[42]. In an independent and ongoing study of iPSC-derived neuronal lineage models and comparisons to mouse tissues, neural stem cells (NSCs) and NGN2-induced neurons (iNs) were derived from these isogenic iPSCs. bRNA-seq was completed on the NSCs and iNs and used for comparisons here. We observed that 9504 genes were expressed in both the NSCs with 16p11.2 deletion and donor-derived cerebral organoids (SetA), of which 93 of these genes (0.98%) were differentially expressed with FDR ≤ 0.05 in both the NSCs and organoids, after excluding the genes in the 16p11.2 locus. In contrast, we observed that 9526 genes were expressed in both the iNs with 16p11.2 deletion and donor-derived cerebral organoids, out of which, none of these genes were differentially expressed with FDR ≤ 0.05 in both the iNs and organoids, after excluding the genes in the 16p11.2 locus. We observed a similar enrichment of differentially expressed genes between the donor-derived cerebral organoids and NSCs with 16p11.2 duplication (OR = 10.4, 95% CI = [5.6, 21.5], Fisher's Exact Test $P < 2.2 \times 10^{-16}$), as well as when using more stringent criteria.

These observations provide further evidence that the differentially expressed genes from the patient-derived cerebral organoids are significantly more similar to the differentially expressed genes from the isogenic NSCs than the isogenic iNs with the same 16p11.2 deletion or duplication than by chance. Transcriptomic alterations in cortical neural progenitor cells from donors with 16p11.2 deletions or duplications were also reported recently[43]. However, it was also previously reported that there was no significant difference in the proliferation of neural progenitor cells from donors with 16p11.2 deletions[43,44].

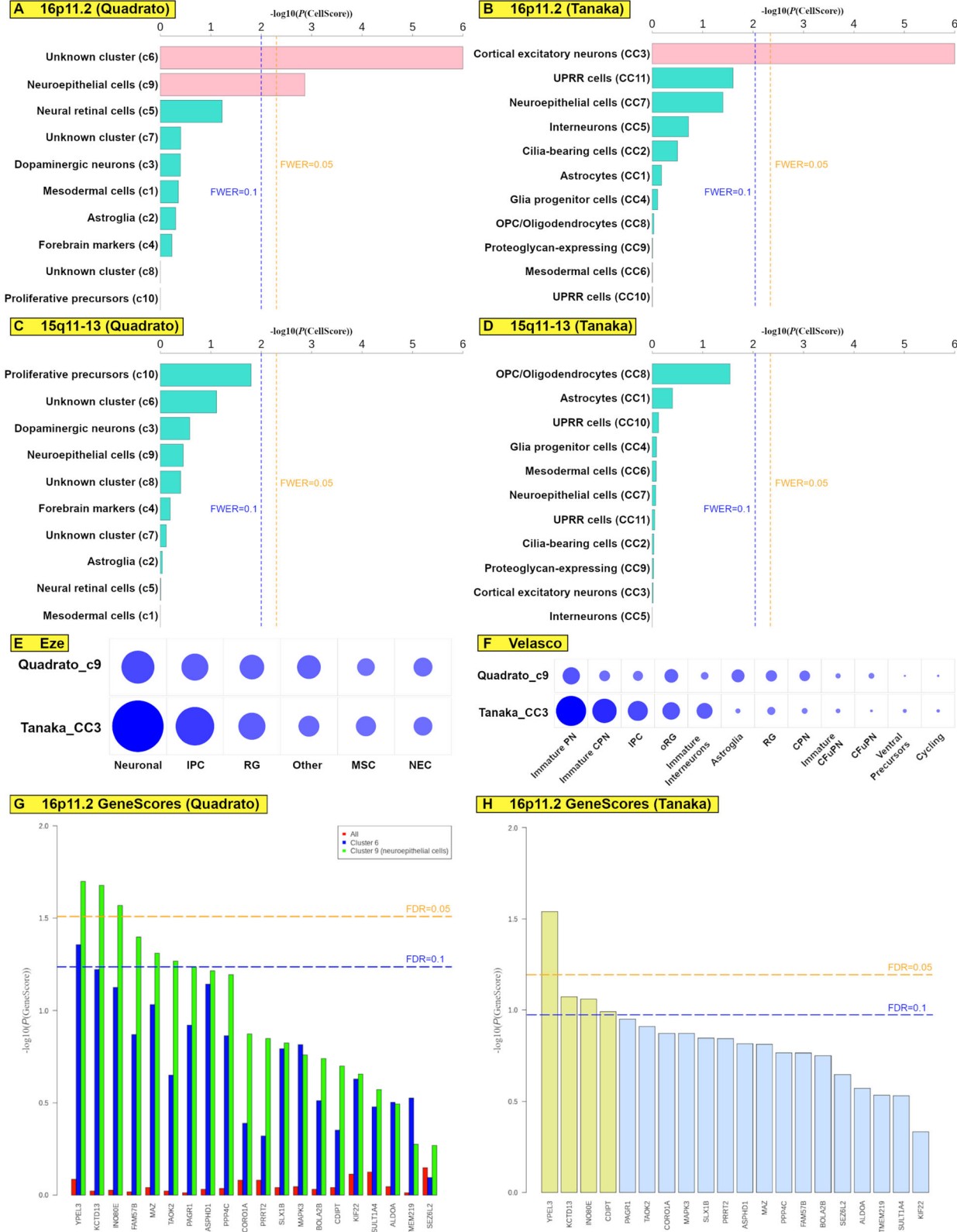

**Fig. 3 Prioritized critical cell types for the 16p11.2 and 15q11-13 loci.** All data were analyzed from 71 bRNA-seq samples across 25 donors. **A** CellScore results with one-sided tests for 16p11.2 (Quadrato dataset[1]); clusters with FWER ≤ 0.1 in pink adjusted for multiple comparisons. **B** CellScore results with one-sided tests for 16p11.2 (Tanaka dataset[19]); clusters with FWER ≤ 0.1 in pink adjusted for multiple comparisons. **C** CellScore results with one-sided tests for 15q11–13 (Quadrato dataset[1]). **D** CellScore results with one-sided tests for 15q11–13 (Tanaka dataset[19]). **E** Fine-mapping identities of critical cell types for 16p11.2 (Eze dataset[20]); sizes of the circles represent mean gene overlaps between cell type clusters. **F** Fine-mapping identities of critical cell types for 16p11.2 (Velasco dataset[13]); sizes of the circles represent mean gene overlaps between cell type clusters. **G** GeneScore results for 16p11.2 (Quadrato dataset[1]). **H** GeneScore results for 16p11.2 (Tanaka dataset[19]).

**Fine-mapping of cell-type identities using large-scale neuro-developmental maps point to the role of immature neurons and intermediate progenitor cells for the 16p11.2 locus**. To ensure consistencies in assigning cell type identities across different studies, and to fine-map the critical cell types more precisely during neurodevelopment, we used two neurodevelopmental maps that were reconstituted from scRNA-seq data on brain organoids and fetal brains. The first neurodevelopmental map by Velasco et al. comprised of 12 cell types[13], and we calculated the percentage overlap among genes from each cell type cluster reported by Quadrato et al. and Tanaka et al.[1,19] (Fig. 3E, Supplementary Data 10). We found that the neuroepithelial cell cluster (c9) from the Quadrato study[1] overlapped most closely with immature projection neurons (mean overlap = 0.37%), and that the unknown cell type (c6) from the Quadrato study[1] overlapped most closely with outer radial glia cells (mean overlap = 0.29%). The CC3 cluster from the Tanaka study[19] overlapped most closely with immature projection neurons in the neurodevelopmental map (mean overlap = 1.14%), similar to the c9 cluster from the Quadrato study[1].

We used a second neurodevelopmental map by Eze et al. that comprised of six cell types[20], and calculated the percentage overlap among the genes from each cell type cluster reported by the Quadrato and Tanaka studies[1,19] (Fig. 3F, Supplementary Data 10). We found that the c9 cluster from the Quadrato study[1] overlapped most closely with the neuronal and intermediate progenitor cell clusters (mean overlaps = 8.1% and 5.4% respectively). Similarly, the CC3 cluster from the Tanaka study[19] overlapped most closely with the neuronal and intermediate progenitor cell clusters (mean overlaps = 19.8% and 11.1% respectively). These results suggest that the critical cell types for the 16p11.2 locus are likely to be immature neurons and intermediate progenitor cells.

To evaluate the degree of independence among the genes in the c9 cluster from the Quadrato study[1] and the CC3 cluster from the Tanaka study[19], we calculated the correlations between the mean overlaps across the two neurodevelopmental maps for both the c9 and CC3 clusters (Supplementary Data 11). We observed high correlations between both the c9 and CC3 clusters using both neurodevelopmental maps ($r = 0.71$, $P = 9.1 \times 10^{-3}$; $r = 0.95$, $P = 3.6 \times 10^{-3}$). However, there were stronger correlations between the CC3 and c5 clusters ($r = 0.96$, $P = 8.7 \times 10^{-7}$; $r = 0.98$, $P = 6.3 \times 10^{-4}$), even though the c5 cluster was not implicated as the critical cell type from the 16p11.2 donor-derived organoids. This indicates that there is likely to be independence among the genes implicating the c9 and CC3 clusters as critical cell types in the 16p11.2 locus.

**Data integration of bRNA-seq data from post-mortem brain samples and scRNA-seq data from control cerebral organoids to identify critical cell types**. A prior publication had performed RNA sequencing on post-mortem brain samples of the cortex that were obtained from nine individuals with 15q11–13 duplications and 49 control individuals[39]. We calculated CellScores for each of the 10 cell type clusters in the Quadrato study[1] using the differential expression results from the post-mortem brain samples, and calculated a weighted average $P$(CellScore) using the results from the patient-derived cerebral organoids and post-mortem brain samples with 15q11–13 duplications (Supplementary Data 12). Similar to our results from the patient-derived cerebral organoids, there were no cell-type clusters identified from the post-mortem brain samples that were significantly perturbed.

**Non-cell type-specific co-transcriptional network modeling cannot prioritize driver genes in 16p11.2 and 15q11-13**. These ASD-associated CNVs are typically large and span across at least 10 genes. Similar to the identification of driver versus passenger genes in cancers, it has been challenging to identify which of the genes in these ASD-associated CNV loci are more likely to be driver genes (genes whose expression would perturb the proportions of the critical cell types). The prioritization of candidate driver genes, or combinations of genes, is important for follow-up studies, for instance, to create knockouts in animal models or organoids for understanding the biological effects of knockouts in these genes[35].

In the 16p11.2 locus, a prominent study using zebrafish identified *KCTD13* as the key causal gene in the locus[36], although other studies have also shown strong evidence for other genes in the locus such as *TAOK2* [NCBI Gene ID: 9344] and *MAPK3* [NCBI Gene ID: 5595][45–47]. CNV analyses on the whole-exome sequence data from one ASD proband with 16p11.2 deletion in our study (14824.x13) found a smaller exonic deletion spanning across exons in *TAOK2* and an intron in *BOLA2B* [NCBI Gene ID: 654483] (Supplementary Data 3).

In the 15q11–13 locus encompassing 11 genes, several studies have identified *UBE3A* as the major causal gene for ASD[48,49]. Although there is supporting evidence for other candidate causal genes such as *CYFIP1* and *HERC2* in the locus[50,51], there is also evidence supporting that *CYFIP1* is not a causal gene in the locus[52]. Whole-exome sequencing on the iPSCs from one of the ASD probands with 15q11–13 duplication (901) and her unaffected mother (902) showed that they harbored a rare stop-gained mutation (p.Q3441X) in *HERC2*, which is one of the genes in the 15q11–13 locus.

One approach to identifying driver genes is to use scRNA-seq for the identification of genes in the CNV loci that are differentially co-expressed in the critical cell types associated with 16p11.2 deletions or 15q11–13 duplications. However, the expression for the genes in the 16p11.2 and 15q11–13 loci range from the 1.8th to 91st percentiles detected from bRNA-seq (Supplementary Data 13), and the expression for most of these genes cannot be detected from sequencing a relatively small number of cells using scRNA-seq[1]. We hypothesized that integration of bRNA-seq data from the patient-derived cerebral organoids with scRNA-seq data from brain organoids and fetal brains can be harnessed to identify candidate driver genes in these CNV loci by quantifying the effects of co-expressed gene perturbations in the cerebral organoids. Our assumption is that candidate driver genes are likely to result in more perturbations in downstream genes than candidate passenger genes.

To identify downstream targets of each gene in an unbiased manner, we first calculated the Pearson's correlations for each of the genes of interest in the CNV loci, with all genes detected from RNA sequencing in the BrainSpan Project, and used the correlations in expression from the BrainSpan Project as a proxy for co-expression connectivity with our genes of interest. Next, we developed a statistical method termed GeneScore, which is a weighted sum of the co-expression connectivity, and the weights are the $-\log_{10}(P$-values) from our differential expression analyses. As a normalization factor, we used the genomic control, which is the ratio of the observed median to the expected median test statistic[53].

Among the 22 genes in the 16p11.2 locus that are expressed in cerebral organoids, 20 of these genes are also expressed in post-mortem brain samples from the BrainSpan Project. When we calculated GeneScores$_{all}$ using all genes detected from the 16p11.2 organoid RNA sequence data, we found that we were unable to prioritize any of the 11 genes in the 16p11.2 locus

($P$(GeneScore$_{all}$) = 0.71 to 0.97, Fig. 3G, Supplementary Data 14). Among the 13 genes in the 15q11–13 locus that are expressed in cerebral organoids, 11 of these genes are also expressed in post-mortem brain samples from the BrainSpan Project. We calculated GeneScores$_{all}$ using all genes detected from the 15q11–13 organoid RNA sequence data, but were unable to prioritize any of the 11 genes in the 15q11–13 locus ($P$(GeneScore$_{all}$) = 0.38 to 0.39, Supplementary Data 14).

**Cell type-specific co-transcriptional network modeling can prioritize driver genes in 16p11.2 and 15q11-13.** Given our earlier observation that c9 cluster from the Quadrato study[1] and CC3 cluster from the Tanaka study[19] are likely to be important for the 16p11.2 locus, we hypothesized that we can obtain higher sensitivity to detect candidate driver genes by focusing on cell-type-specific signatures. When we adapted our GeneScore calculations to include only cell-type-specific genes that were identified in the Quadrato c9 cluster[1], we found that three genes (*YPEL3*, *KCTD13,* and *INO80E*) were significantly prioritized as high-confidence candidate driver genes with FDR ≤ 0.05 (Fig. 3G, Supplementary Data 14, 15), and another four genes (*FAM57B* [NCBI Gene ID: 83723], *MAZ* [NCBI Gene ID: 4150], *TAOK2* and *PAGR1* [NCBI Gene ID: 79447]) were prioritized as lower confidence candidate driver genes with FDR ≤ 0.1 in c9. Interestingly, we did not find any high-confidence candidate gene with FDR ≤ 0.05 in c6, and only one lower confidence candidate gene with FDR ≤ 0.1 in c6 (*YPEL3*).

To replicate our results for the 16p11.2 locus, we calculated GeneScores using the cell-type-specific genes in the CC3 cluster from the Tanaka study[19]. *YPEL3* was similarly prioritized as a high-confidence candidate driver gene at FDR ≤ 0.05, and both *KCTD13* and *INO80E* were prioritized at FDR ≤ 0.1 (Fig. 3H, Supplementary Data 14, 15). Another gene (*CDIPT* [NCBI Gene ID: 10423]) was also prioritized at FDR ≤ 0.1.

One of the three high-confidence driver genes in the c9 cluster (*KCTD13*) was initially implicated as a gene that modulated brain sizes in zebrafish[36], but other studies using KCTD13-deficient mice and zebrafish did not observe any differences in brain sizes or neurogenesis[37,38]. Through the Orgo-Seq framework on patient-derived cerebral organoids, we found that *KCTD13* is one of the three genes in the 16p11.2 locus that appears to modulate the proportions of immature neurons in human cerebral organoids. It was also reported that KCTD13-deficient mice and zebrafish had increased levels of RHOA expression, and that RhoA might be a therapeutic target for disorders associated with *KCTD13* deletion[37]. However, we did not observe any difference in RHOA expression from the patient-derived cerebral organoids (fold change = 1.01 for SetA, FDR = 0.21), suggesting that 16p11.2 deletions in human cerebral organoids might be perturbing a RhoA-independent pathway, or that RHOA expression is perturbed only among a subset of cell types within the cerebral organoids. Similarly, a recent publication reported that inhibitors of RhoA signaling did not rescue deficiencies observed in KCTD13-knockout neurons[54].

**Deletions in *KCTD13* and 16p11.2 similarly impact the S-phase of cell cycle division.** Recent research reported that isogenic KCTD13-deficient neural progenitor cells have a significantly lower percentage of cells in the S-phase of the cell cycle compared to wild-type neural progenitor cells[54]. To evaluate if the RNA sequence data from our patient-derived cerebral organoids comparing cases with 16p11.2 deletions versus controls without the deletions (SetA) can similarly reflect an enrichment of transcriptomic perturbations in the S-phase, we performed gene ontology (GO) enrichment on the list of significantly

differentially expressed genes from SetA. Gene ontology enrichment analyses of eight different GO terms involved in cell division, proliferation, and replication showed that the differentially expressed genes in SetA were most likely to be involved in cell division (FDR = $2.3 \times 10^{-10}$). There were three GO terms involved in the cell cycle with FDR ≤ 0.05, and the differentially expressed genes from the patient-derived cerebral organoids were most significantly enriched for the G1/S transition of the mitotic cell cycle term (FDR = $5.5 \times 10^{-7}$) compared to the G2/M transition (FDR = $2.2 \times 10^{-2}$) and the mitotic spindle assembly checkpoint (FDR = $3.8 \times 10^{-4}$). These results show that the transcriptomic perturbations in the patient-derived cerebral organoids with 16p11.2 deletions are similar to the transcriptomic perturbations found in neural progenitor cells with *KCTD13* deletions, and this provides another line of evidence that *KCTD13* is one of the driver genes in the 16p11.2 locus.

**Evidence for the role of multiple driver genes in the 16p11.2 locus.** It has been of great interest if there is a single driver gene in the 16p11.2 locus, as previously reported[36], or if multiple driver genes in the 16p11.2 locus can contribute to the ASD-associated molecular signatures or phenotypes observed[46,55]. A previous publication reported that 13 cell cycle-associated genes were expressed at significantly lower levels in KCTD13-deficient NPCs compared to wild-type NPCs. None of these 13 genes are expressed at significantly lower levels in the patient organoids with 16p11.2 deletions compared to control organoids. The results show that KCTD13-deletions in human NPCs are insufficient to recapitulate the full transcriptomic perturbations observed in the donor-derived organoids with 16p11.2 deletions. Moreover, in addition to *KCTD13*, Orgo-Seq prioritizes two other candidate driver genes in the 16p11.2 locus (*YPEL3* and *INO80E*).

## Discussion

To-date, there are over a hundred genes and loci associated with complex neuropsychiatric disorders such as ASD[25,56,57]. Cerebral organoids are an emerging human-derived model system for identifying cell types and cell type-specific processes that are perturbed by genetic variation associated with complex neurodevelopmental and neuropsychiatric disorders[2,3,8,13,58,59]. These cerebral organoids comprise of many different cell types, so this effectively allows us to test multiple hypotheses in multiple cell types that were differentiated under the same conditions. It is interesting to note that we were unable to prioritize any candidate driver genes when using co-expression patterns of all genes whose expressions were detected in the cerebral organoids, but we were able to nominate genes that appeared to drive co-expression patterns within specific cell types, emphasizing the power of evaluating cell type specificity[60]. These approaches will become increasingly valuable in cross-disorder studies where etiological overlap has been identified, such as in neuropsychiatric disorders. Cerebral organoids can be powerful model systems to evaluate cell-type-specific commonalities in disease processes using a genotype-driven approach.

A major strength of using donor-derived organoids for discoveries is that the donor-derived organoids can model the diverse genetic backgrounds found in humans, and overcome some of the limitations faced with using isogenic iPSC derivatives or inbred animal models. As such, it will be increasingly important to develop technologies and methods that enable unbiased high-throughput discoveries using donor-derived organoids, to leverage on the unperturbed complexity of human genetics for making important discoveries in disease biology[61].

In our work, we describe the Orgo-Seq framework to allow the identification of cell types and cell type-specific driver genes from

donor-derived cerebral organoids that are important in ASD-associated CNVs such as 16p11.2 and 15q11–13, by integrating multi-transcriptomics data (bRNA-seq and scRNA-seq) from multiple sources (cerebral organoids and post-mortem brains). Orgo-Seq allows us to overcome technical limitations such as capture efficiencies with detecting critical cell types and cell type-specific driver genes using scRNA-seq alone, but leverage the strengths of scRNA-seq such as the unbiased discovery of critical cell types from a mixture of cell types. The framework can be generalized for identifying specific types of neurons or glia cells[62], as well as cell type-specific driver genes for many other CNVs that have been robustly associated with complex neurodevelopmental and neuropsychiatric disorders[16,63].

In addition, as high-quality scRNA-seq data are generated from increasingly large numbers of single cells[64], or scRNA-seq data are generated using recent spatial-informative technologies[65,66], the Orgo-Seq framework allows us to integrate additional scRNA-seq data with the bRNA-seq data that we had already generated from our donor-derived organoids to make discoveries about cell types and cell type-specific driver genes. The framework can also be generalized for identifying cell types and cell type-specific driver genes using bRNA-seq data that had been generated from human post-mortem brains, without the need to perform scRNA-seq directly on post-mortem brain samples with limited availability.

In our current study, we found that we were able to observe transcriptomic differences that can shed insights into the critical cell types and cell type-specific processes that are important in neurodevelopmental and neuropsychiatric disorders such as ASD using early 46-day-old cerebral organoids. Prior work demonstrated that even in these early 1–2 month-old cerebral organoids, there are robust transcriptomic and cellular differences that could be detected for neurodegenerative diseases such as Alzheimer's Disease[67,68]. It will be interesting use Orgo-Seq to integrate additional scRNA-seq data from human post-mortem brain tissue across different developmental timepoints to obtain insights into the disease biology of neurodevelopmental and neurodegenerative diseases.

In summary, we have established a quantitative framework for generating and validating hypotheses about cell type-specific driver genes involved in complex neurological disorders using a human-derived model system. As a future direction, it will be exciting to explore the possibility of developing a precision medicine framework to rapidly identify critical cell types and cell type-specific driver genes in individual donors, and the framework can complement DNA sequencing to enable the identification of putative causal cell types and cell-type-specific genes and gene networks in an individual patient for personalized diagnostics.

## Methods

**Standard protocol approval**. Research performed on samples and data of human origin was conducted according to protocols approved by the institutional review boards of Harvard Medical School and UMass Chan Medical School.

**Donor samples**. A total of 25 iPSCs (1 clone per iPSC) were obtained as from the Personal Genome Project, Coriell Institute, ATCC, Harvard Stem Cell Institute and Simons VIP collection[43,44] (Table 1). Informed consent was obtained from all donors through the various biobanks.
Simons VIP Collection:
http://simonsfoundation.s3.amazonaws.com/share/Policies_and_forms/2014/svip/SVIPSampleConsent9-13-12.pdf
Coriell: https://www.coriell.org/0/sections/Support/NIGMS/Model.aspx?PgId=216)
ATCC: https://www.atcc.org/support/order-support/permits-and-restrictions
Personal Genome Project: https://pgp.med.harvard.edu/how-it-works
All iPSCs and cerebral organoids were tested negative for mycoplasma using the LookOut Mycoplasma PCR Detection kit (Sigma MP0035). All iPSCs except for

PGP1 were validated and characterized by Coriell Institute (karyotyping, embryoid body formation and PluriTest), ATCC (karyotyping, antigen expression of SSEA4/TRA-1–60 and SSEA1), Harvard Stem Cell Institute (karyotyping) or Simons VIP collection (single nucleotide polymorphism microarray). We performed flow cytometry (CytoFlex LX) to confirm that >90% of the iPSCs from each donor are positive for TRA-1–60 (Novus Biologicals NB100-730F488). If we had observed donor iPSCs with <90% TRA-1-60$^+$ cells, we typically perform an anti-TRA-1–60 bead purification step (Miltenyi Biotec 130-100-832) before re-testing with flow cytometry. It will be interesting to compare the results from multiple clones from the same donors to understand the clonal variability across many donors[5,43,44].

**CNV analyses**. iPSCs from all donors were passaged until they were confluent, and 2 million cells per donor were counted using an automated cell counter, and washed twice in 1x DPBS, before flash freezing the cell pellets. The frozen cell pellets were sent on dry ice to Cell Line Genetics, where genomic DNA was extracted from the cells, and quality control was performed using Nanodrop, Qubit, and agarose gel analyses. The Agilent 60k standard aCGH was used to identify CNVs, and the CNVs were compared to the Database of Genomic Variants CNV-DGV_hg19_May2016 (http://dgv.tcag.ca/dgv/app/home) to identify CNVs that are common in the general population (Supplementary Data 2). All four donors with 15q11–13 duplications were confirmed to harbor the duplications, all nine donors with 16p11.2 deletions were confirmed to harbor the deletions, and all 12 control individuals were confirmed not to harbor any duplication in the 15q11–13 locus, or deletion in the 16p11.2 locus.

To identify smaller exonic CNVs, we further performed CNV analyses from whole-exome sequence data on all donor iPSCs. DNA was extracted from iPSC cell pellets for all donors using the standard protocol for AccuPrep Genomic DNA Extraction Kit (Bioneer K-3032), and we evaluated the quantity and quality of the extracted DNA samples using Nanodrop. 1µg of DNA per iPSC was sent on dry ice to Macrogen, where quality control was performed using Quant-iT PicoGreen dsDNA Assay Kit (Life Technologies P7589) with Victor X2 fluorometry, and the Genomic DNA ScreenTape assay (Supplementary Data 1). The DNA Integrity Number (DIN) threshold used for exome sequencing was 6, and the mean DIN across all control samples was 8, the mean DIN across all samples with 15q11–13 duplications was 7.7 and the mean DIN across all samples with 16p11.2 deletions was 7.9, but there were no significant differences between the DNA quality from the iPSCs with 15q11–13 duplications versus the control iPSCs (two-sided Wilcoxon $P = 0.2$), or the iPSCs with 16p11.2 deletions versus the control iPSCs (two-sided Wilcoxon $P = 0.62$). The Agilent SureSelect V5-post kit was used for capture and the library was sequenced using NovaSeq 6000 (150 paired end). CNV calling on the exome sequence data was performed using CoNIFER v0.2.2[69], and all exonic CNVs detected from the iPSCs are shown in Supplementary Data 3. Among the cases with 15q11–13 duplications or 16p11.2 deletions, only a smaller deletion in the 16p11.2 locus encompassing exons in *TAOK2* and an intron in *BOLA2B* was found detected from the whole-exome sequence data for proband 14824.x13. Whole-genome sequencing on all samples with 16p11.2 deletions or 15q11–13 duplications was performed at Macrogen to detect the breakpoints of the deletions or duplications, and CNV calling was performed using CNVnator v0.4.1[70] (Supplementary Data 4).

**Cerebral organoid differentiation**. We adapted our cerebral organoid differentiation protocol according to a previously described protocol[2] (Supplementary Fig. 1A). For embryoid body formation, cells were counted using an automated cell counter and 900,000 iPSCs were re-suspended in 15 ml of mTeSR medium (Stemcell Technologies 85850) with 50 µM Y-27632 dihydrochloride monohydrate (Santa Cruz sc-216067A), and 150 µl was seeded into individual wells of a 96-well ultra-low attachment Corning plate (ThermoFisher CLS7007). On Day 6, 50 µl of mTeSR medium with a single embryoid body was transferred to individual wells of 24-well ultra-low attachment Corning plates (ThermoFisher CLS3473) with 500 µl of neural induction media per well. On Day 8, another 500 µl of neural induction media was added to each well of the 24-well plates. On Day 10, a droplet comprising of 10 µl of neural induction media with an organoid was placed onto a single dimple on Parafilm substrate, and 40 µl of Matrigel (Corning 354234) was added to each organoid to encapsulate it. The Matrigel droplets were incubated at 37 °C for 15 min before they were scraped into single wells of the 24-well plates using a cell scraper. One milliliter of differentiation media with 10% penicillin streptomycin (ThermoFisher 15140122) per well was used to passage the organoids every 2–4 days, and the plates of organoids were placed on an orbital shaker at 90 rpm in the incubator. A previous publication noted that a bioreactor-related growth environment is a key factor in controlling cell-type identity from organoids to organoids[1], and similarly, we had observed batch effects in the rates of cell death while differentiating multiple organoids in the same well of multi-well plates. To reduce batch effects and biases in cell-type compositions that were previously reported in individual organoids[1], we differentiated individual organoids in single wells of 24-well plates on an orbital shaker.

**Cerebral organoid cryosection and immunostaining**. Cerebral organoids were rinsed twice with 1× DPBS, fixed in 4% paraformaldehyde at 4 °C for 30–60 min, immersed in 30% sucrose overnight, and embedded in optimal cutting temperature

compound, and 8-micron sections are collected with a cryostat. Cryosections of fixed cerebral organoids were immunostained with antibodies against Sox2 (Santa Cruz sc-17320, 1:200 dilution), Tbr2 (Abcam ab-23345, 1:200 dilution), Tuj1 (Covance MMS-435P, 1:1000 dilution) and Alexa Fluor secondary antibodies (ThermoFisher).

**RNA extraction, sequencing, alignment, and annotation**. It was previously noted that some cell types are found in only 32–53% of organoids, using scRNA-seq[1]. In order to reduce variability across replicates, as well as to obtain sufficient representation of all cell types, we pooled 20 separate organoids from different wells and different plates, as one replicate. The organoids in each replicate were pelleted at $1000\,g$ for 1 min, and the supernatant was removed, before washing twice in DPBS. RNA from 1 to 3 replicates was extracted for each individual (Supplementary Data 1). The organoids were homogenized using mechanical disruption in lysis buffer, and RNA extraction was performed using the PureLink RNA Mini Kit (ThermoFisher 12183018 A), according to the manufacturer's protocol. RNA samples were treated with Ambion DNase I (ThermoFisher AM2222) according to the manufacturer's protocol, before they were frozen and sent on dry ice to Macrogen.

At Macrogen, DNA quantity was measured using Quant-iT PicoGreen dsDNA Assay Kit (Life Technologies P7589) with Victor X2 fluorometry, and RNA quantity was measured using Quant-iT RiboGreen RNA Assay Kit (Life Technologies R11490). The RNA Integrity Number (RIN) was measured using an Agilent Technologies 2100 Bioanalyzer or TapeStation, and the RIN value threshold used was 6 (Supplementary Data 1). Ribosomal RNA depletion using TruSeq Stranded RNA with Ribo-Zero (Human) and paired-end 101 bp sequencing with at least 30 million reads per sample was performed. Library size checks were performed using an Agilent Technologies 2100 Bioanalyzer or TapeStation, and quantification of the libraries was performed according to the Illumina qPCR quantification guide. Reads were trimmed using Trimmomatic v0.32[71], then mapped to the hg19 human genome sequence using TopHat v2.0.13[72], and transcript assembly was performed using Cufflinks v2.2.1[73] to calculate the fragments per kilobase per million reads (FPKM) values for each transcript. In addition, the reads were mapped to the hg19 sequence using STAR v2.4.0f1[74], and a single nucleotide variant calling on the aligned sequences was performed using GATK v3.3-0 HaplotypeCaller[75]. Annotation for the single nucleotide variants was performed using SeattleSeq Annotation 138 (https://snp.gs. washington.edu/SeattleSeqAnnotation138/)[76], and single nucleotide variants detected from the RNA sequence data were compared between replicates from the same individual and verified for concordance ($r > 0.95$), to ensure that there was no sample mix-up.

**Data processing and quality control**. The mean RIN values for the control samples, 15q11–13 samples, and 16p11.2 samples were 7.9, 8.1, and 8.2 respectively (Supplementary Data 1). We performed a two-sided Wilcoxon rank-sum test between the RIN values for the control samples versus the 15q11–13 samples, but did not observe significant differences ($P = 0.43$). Similarly, we did not observe significant differences between the RIN values for the control samples versus the 16p11.2 samples ($P = 0.13$). Neither did we observe significant differences between the RIN values for the 15q11–13 samples versus the 16p11.2 samples ($P = 0.47$).

After selecting the transcript with the highest mean FPKM across all samples (including all cases and controls) for each gene, there were 25,727 unique transcripts or genes. We further performed quality control to remove genes that were not expressed, or had high intra-individual or inter-individual variance. Genes that were not expressed in the cerebral organoids (mean FPKMs across all samples <2) were removed, resulting in a smaller set of 11,300 genes. We calculated the mean FPKMs across all samples, including all case and control samples. However, we used only the control samples for calculating the standard deviations in gene expression, to preserve genes that truly contribute to biological variation between the case and control organoids. Inverse rank-sum normalization was performed on the expression values that were subsequently used in the downstream analyses, as the normalization procedure reduces outlier expression values. To test for Sendai virus clearance, we used a list of 10 most highly induced genes upon Sendai virus infection reported by Mandhana and Horvath[77], and found that none of these 10 genes were expressed in our samples with mean FPKM ≥ 2.

With every technology or system, there are some measurements that will be made below the background noise, or below the technical sensitivity of the system. These measurements are usually not relied upon because there is low confidence in the accuracy of the measurements. Similarly, we identified some genes from the bRNA-seq data that are highly variable in their expression, and we cannot confidently estimate the expression of these genes using our system. There were 860 genes with more than 2 standard deviations in any intra-individual variance calculated across the control samples (Supplementary Figs. 2 and 3A, Supplementary Data 5), and 869 genes with more than 1.5 standard deviations in inter-individual variance calculated between the control samples (Supplementary Figs. 2 and 3B, Supplementary Data 5), resulting in a total of 1322 unique outlier genes. After removing all outlier genes with high variability, there are a total of 9978 unique genes. Pairwise Pearson's correlations ($r^2$) were performed for each pair of replicates from an individual to calculate the intra-individual correlations, and each pair of replicates from different individuals to calculate the inter-

individual correlations. Variability in cell-type compositions across our samples was reduced by ensuring that only genes with low intra-individual or inter-individual variability were included in our analyses.

**Comparing BrainSpan samples with cerebral organoid samples**. The BrainSpan project (http://www.brainspan.org) provides a high-resolution map of 22,326 genes detected using RNA sequencing on 578 post-mortem brain samples from various brain regions in prenatal brains (8 pcw) to adult brains (40 years old)[21]. We downloaded the "RNA-Seq Gencode v10 summarized to genes" dataset from the BrainSpan Project for our analyses (http://www.brainspan.org/static/download. html). For comparing RNA sequence data from prenatal brain samples from the BrainSpan Project with RNA sequence data from cerebral organoids, we included only brain regions where more than 50% of samples were available for those regions (≥9 samples). We performed a two-sided Wilcoxon rank-sum test to evaluate if the mean Pearson's correlations between the organoids and prenatal brain samples were significantly higher than the mean Pearson's correlations between the organoids and postnatal brain samples. We further calculated Pearson's correlations for each pair of genes from the BrainSpan RNA sequence data.

We observed that after removing highly variable genes, the Pearson's correlations between RNA sequence data from the organoids with the 578 post-mortem brain samples ranged from 0.23 to 0.74, using only one technical replicate from each of the 12 control donors (Supplementary Fig. 7A). Prior to removing highly variable genes, there was a larger variance (Supplementary Fig. 7B), that was primarily driven by high outlier correlations in all replicates from two control samples. For instance, the mean Pearson's correlation between all organoid samples excluding the outliers, with cerebellar cortex from a 16pcw fetal brain sample is 0.33. However, the mean Pearson's correlations between the 16pcw fetal brain sample with organoid samples are 0.84 and 0.92 with the outlier samples.

**Comparisons of highly variable genes with genetic and genomic associations for ASD**. We compared the genes with high intra-individual or high inter-individual variability with previously reported ASD-associated genes from two publications[24,25]. There are 102 genes associated with ASD reported in Satterstrom et al.[25], and we found that five of these genes (4.9%) had high intra-individual variability and five of them (4.9%) had high inter-individual variability in the bulk RNA sequence data from the organoids. Similarly, for Gandal et al.[24], there are 1,611 genes associated with ASD and 44 of these genes (2.7%) had high intra-individual variability, and 61 of these genes (3.8%) had high inter-individual variability in the bulk RNA sequence data from the organoids. These observations show that the genes with high inter-individual or intra-individual variability in RNA sequence data from the cerebral organoids are not enriched for genes with genetic or genomic associations with ASD.

**Comparisons of bulk RNA sequence data from cerebral organoids with post-mortem brain samples from the BrainSpan Project**. To determine how accurately RNA sequence data from cerebral organoids can reflect RNA sequence data from post-mortem human brain samples, we compared our data with data from 578 post-mortem brain samples in the BrainSpan Project[21] (Supplementary Fig. 6A–F). We observed a wide range of correlations (Pearson's $r^2$ from 0.21 to 0.82, Supplementary Fig. 6A). The mean correlations between the organoids with individual prenatal brain samples (maximum $r^2 = 0.67$, minimum $r^2 = 0.38$) were significantly higher than the mean correlations between the organoids with individual postnatal brain samples (maximum $r^2 = 0.58$, minimum $r^2 = 0.28$), shown in Supplementary Fig. 6B (two-sided Wilcoxon $P < 2.2 \times 10^{-16}$).

The highest mean correlations were with prenatal brain samples from regions such as the hippocampus ($r^2 = 0.61$), primary visual cortex ($r^2 = 0.61$) and amygdala ($r^2 = 0.61$), shown in Supplementary Fig. 6C. The lowest correlations with prenatal brain samples were across brain regions such as the mediodorsal thalamus ($r^2 = 0.49$), ventrolateral prefrontal cortex ($r^2 = 0.56$) and primary somatosensory ($r^2 = 0.56$). Similar observations were made when comparing RNA sequence data from cerebral organoids prior to removing the outlier genes (Supplementary Fig. 6D–F).

**Differential gene expression analyses**. We used variancePartition to identify potential drivers of variation in the RNA sequence data from the organoids[27], and found that most of the variation in the data was unaccounted for by eight sample variables: ethnicity, sex, age, the origin of sample used for iPSC reprogramming, type of reprogramming, center that distributed the iPSC line, ASD diagnosis and CNV genotype (Supplementary Fig. 4A). Gene ontology enrichment of the genes with >99% variance explained by the residuals showed that these genes are enriched in the mitochondrial envelope (Supplementary Fig. 4B).

Principal components analyses on all case and control samples showed that most of the variance in gene expression (88%) could be accounted for by the first principal component (PC1) alone (Supplementary Fig. 5A). We further observed that age, the origin of sample, and the type of reprogramming are significantly correlated with PC1 alone, but not with the second and third principal components (Supplementary Fig. 6). These results suggest that PC1 is a surrogate variable for age, the origin of sample, and type of reprogramming, and subsequently, we included PC1 as a covariate in the differential expression analyses.

We performed differential expression analyses using linear regression in R (lm function), with PC1 as a covariate, and performed multiple hypotheses correction using the Benjamini-Hochberg false discovery rate in R (p.adjust). To identify the sources of variation in the expression data, we performed variancePartition using the default parameters in the documentation[27]. Given the relatively small number of samples used in our study[78], and since PC1 captures 88% of the variance in gene expression and is a surrogate factor for several sample variables, we included only PC1 as a covariate in our linear regression analyses to identify differentially expressed genes. We further plotted the first two principal components between control individuals without deletions or duplications, and individuals with 16p11.2 deletions or 15q11–13 duplications, but did not observe major stratification between the cases and controls in the first two principal components (Supplementary Fig. 8).

Given that the inter-individual correlations observed between samples from different individuals are similarly high compared to the intra-individual correlations observed between replicates from the same individual, and given the relatively small number of individuals in our study that limits the number of permutations, we performed linear regression using all samples as independent samples. We had also performed bRNA-seq on 1–3 replicates for each individual, to ensure that the results were not skewed by RNA sequence data from a few outlier individuals.

**Power calculations for the SetD analyses**. To calculate the power for the SetD analyses, we simulated a normal distribution with mean FPKM gene expression values ranging from 2 to 5 in resilient individuals ($n = 14$ replicates), and mean fold change in individuals with ASD ranging from 1.2–4 ($n = 9$ replicates), and standard deviation = 18.5 (the observed mean), for 1000 times, and calculated the percentage of times we observed an FWER of 0.05 or less in the simulated data. FWER is defined using Bonferroni correction as 0.05/number of genes.

**Comparison of differentially expressed genes from 16p11.2 deletion and 15q11-13 duplication cerebral organoids reveals 9 genes in common**. We compared the differentially expressed genes with FDR ≤ 0.05 between the 16p11.2 deletion SetA and 15q11–13 duplication results, and observed that there were eight genes that were differentially expressed in the same direction for 16p11.2 deletions (SetA) and 15q11–13 duplications (*RPS14, PCDHGB6, TUBGCP5, CYFIP1, ELAVL2, SNHG5, NAP1L5,* and *MYL6B,* Supplementary Data 16). There was a high correlation between the fold changes of these nine genes between 16p11.2 deletions and 15q11–13 duplications (Pearson's $r = 0.92$, $P = 0.0014$).

Another gene product (*HERC2*) was also differentially expressed for 16p11.2 deletions (SetA) and 15q11–13 duplications but in opposite directions. *HERC2* was over-expressed in 15q11–13 duplications cases compared to controls (fold change = 1.48), whereas *HERC2* was under-expressed in 16p11.2 deletion cases compared to controls (fold change = 0.9). Of the nine genes that were differentially expressed, three of them (*TUBGCP5, CYFIP1,* and *HERC2*) were found in the 15q11–13 locus. These results suggest that there are shared key genes that are perturbed by 16p11.2 deletions and 15q11–13 duplications.

There were six genes that were differentially expressed in the same direction for 16p11.2 deletions (SetP) and 15q11–13 duplications (*RPS14, PCDHGB6, ELAVL2, SNHG5, CTNNA2,* and *NAP1L5,* Supplementary Data 16). There was a moderate correlation between the fold changes of these 6 genes between 16p11.2 deletions (SetP) and 15q11–13 duplications (Pearson's $r = 0.73$, $P = 0.064$).

*HERC2* was also differentially expressed for 16p11.2 deletions (SetP) and 15q11–13 duplications but in opposite directions (fold change = 0.8 in 16p11.2 SetP and fold change = 1.48 in 15q11–13).

There were no significantly differentially expressed genes with FDR ≤ 0.05 from the 16p11.2 deletion (SetD) analyses.

**Power calculations for the second hit outside the 16p11.2 locus contributing to ASD risk**. Many risk loci associated with complex disorders such as ASD have incomplete penetrance[79], and are present in control individuals who are not clinically diagnosed with these disorders, albeit at a lower prevalence than in affected individuals. A two-hit model was previously described for another microdeletion locus in 16p12.1, where the microdeletion was reported to exacerbate neurodevelopmental phenotypes in association with other large CNVs[80]. We asked if we could similarly detect a second genetic factor that contributes to increased risk for ASD, in addition to the 16p11.2 deletion background, and this might explain for the incomplete penetrance observed with the 16p11.2 deletions. The SetD comparison between individuals with ASD and 16p11.2 deletions versus resilient individuals with 16p11.2 deletions did not yield any gene with FDR ≤ 0.05. There is <80% power to detect a gene with small effects (fold changes of <4) at FWER≤0.05 given our current sample sizes (Supplementary Data 6).

**Permutation schemes for 16p11.2 SetA and 15q11-13**. We permuted the case-control status of each organoid replicate to obtain null distributions. However, given the relatively small numbers of samples, we wanted to avoid creating permuted instances where the permuted cases are actual case samples and the permuted controls are actual control samples. Differential expression analyses on these permuted instances will result in the detection of true biological differences, instead

of creating a baseline non-biological measurement for the null distribution. As such, we developed a permutation strategy by sampling permuted case samples from the actual control samples only (Supplementary Fig. 9). Furthermore, to ensure that we have the same numbers of case and control samples in our permutations, as the numbers of case and control samples from our actual experiments, we assigned all the actual case samples to be permuted control samples. We refer to the cases in the permutations as "pseudo-cases", and the controls in the permutations as "pseudo-controls".

For 16p11.2 SetA, we performed differential expression analyses for 23 samples differentiated from all individuals with 16p11.2 deletions (cases) versus 36 samples differentiated from unaffected controls without the deletion (controls). To obtain a null distribution, we randomly assigned 23 samples from the 36 control samples as pseudo-cases, and assigned the initial 23 samples, together with the remaining control samples as pseudo-controls, for 1 million permutations to calculate CellScores, and 100,000 permutations to calculate GeneScores. Subsequently, we performed linear regressions with PC1 as a covariate on all the expression data for the permutations.

For the 15q11–13 results, we performed differential expression analyses for 12 samples differentiated from individuals with ASD and 15q11–13 duplications (cases) versus 36 samples differentiated from unaffected controls without the duplications (controls). To obtain a null distribution for comparing the observed statistics, we randomly assigned 12 samples from the 36 control samples as pseudo-cases, and assigned the initial 12 samples, together with the remaining control samples, as pseudo-controls, for 100,000 permutations to calculate CellScores. Subsequently, we performed linear regressions with PC1 as a covariate on all the expression data for the permutations.

**Calculation of CellScore and $P$(CellScore)**. In the Quadrato study[1], there are 10 major clusters of cell types identified using unbiased clustering on scRNA-seq data from cerebral organoids, and each cell cluster has an associated list of genes identified using Drop-seq, and was assigned a cell cluster identity using previously published data from homogeneous cell populations[1]. We observed that in these full lists of cluster genes, there are some genes that are present in multiple cell clusters, and that these genes are not cell type specific. To enrich for cell type specific genes, we further identified a smaller subset of genes that are uniquely found in each cell type cluster but are not present in other cell type clusters, which we termed as "cell-type-specific genes" (Supplementary Data 7). We termed the genes that are found in multiple cell clusters as "non-cell type-specific genes".

In the Tanaka study[19], there are 24 major clusters of cell types identified using unbiased clustering on scRNA-seq data from brain organoids and fetal brains. We calculated the overlap among genes from each of the 24 clusters and identified 11 unique cell-type clusters (Supplementary Data 9). Similarly, we identified a subset of genes as cell type specific genes and non-cell type specific genes (Supplementary Data 7).

We calculated CellScore for each cluster by summing up the $-\log_{10}$-transformed $P$-values from the differential expression results for each gene $y$ in the cluster ($P_y$), divided by the total number of genes in the cluster (Num$_y$), and obtained the difference between the calculated CellScores for the specific genes versus the non-specific genes, where $P_{specific\_y}$ is the $P$-value of each cell type-specific gene in the cluster, Num$_{specific\_y}$ is the number of cell type-specific genes in the cluster, $P_{non-specific\_y}$ is the $P$-value of each non-cell type-specific gene in the cluster, and Num$_{non-specific\_y}$ is the number of non-cell type-specific genes in the cluster. Taking the difference between the calculated CellScores for the cell type-specific genes versus the non-cell type specific genes allows us to obtain a normalized CellScore that is adjusted for other inherent factors that can similarly affect the expression of non-cell type-specific genes.

$$\text{CellScore} = \sum_{\text{all specific\_y}} \frac{-\log_{10} P_{specific\_y}}{\text{Num}_{specific\_y}} - \sum_{\text{all non\_specific\_y}} \frac{-\log_{10} P_{non-specific\_y}}{\text{Num}_{non-specific\_y}} \quad (1)$$

We obtained a null distribution for CellScore by performing 100,000 permutations (see Permutation schemes for 15q11–13 and 16p11.2 SetA), and performed linear regressions for each permutation. Next, we estimated the probability of the observed CellScore for each cluster by comparing with the null distribution (CellScore$_{permuted}$):

$$P(\text{CellScore}) = P(\text{CellScore}_{permuted} \geq \text{CellScore}) \quad (2)$$

To identify significant clusters, we calculated an FWER threshold of 0.05 after Bonferroni correction for multiple hypotheses, i.e., $P = 0.005$; and similarly, for an FWER threshold of 0.1, or $P = 0.01$.

**DNA transfection and single-cell isolation by FACS for isogenic 16p11.2 deletion/duplication iPSC lines**. Transfections were performed using the used Human Stem Cell Nucleofector Kit 1 (Lonza) and Amaxa Nucleofection II device (Lonza) with programs B-016, according to the manufacturer's instructions. After nucleofection, the iPSCs were cultured on Matrigel-coated wells using Essential 8 medium (Invitrogen) supplemented with 10 μM Y-27632 dihydrochloride monohydrate (Santa Cruz Biotech). For subsequent puromycin selection, iPSCs were harvested 24 h after nucleofection in fresh Essential 8 medium with puromycin (0.1 μg/mL). To obtain isogenic iPSC colonies following CRISPR/Cas9 treatment,

single cells were isolated by FACS. At 72 h after nucleofection, the iPSCs were dissociated into a single-cell suspension with accutase (Stemcell Technologies) and resuspended in PBS with 10 μM Y-27632 dihydrochloride monohydrate (Santa Cruz Biotech). All samples were filtered through 5-mL polystyrene tubes with 35-μm mesh cell strainer caps (BD Falcon 352235) immediately before being sorted. After adding the viability dye TO-PRO-3 (Invitrogen), the GFP+ /TO-PRO-3– iPSCs were sorted using BD FACSAriaII with a 100-μm nozzle under sterile conditions and plated into 96-well plates (one cell per well). Once individual iPSC colonies were established (~10–14 days after sorting), cells were passaged and then harvested using a Quick-96 DNA kit (Zymo) and genotyped using both custom PCR primers targeting each deletion breakpoint[42] and ddPCR-based probes as a means of orthogonal genotyping and confirmation of clonality.

**Neural stem cell (NSC) differentiation**. After genotypes of individual iPSC clones were determined, they were expanded and underwent anti-TRA-1-60 selection using magnetic-activated cell sorting (MACS) to select for pluripotent cells (Miltenyi Biotec). TRA-1-60+ cells within two passages of selection were differentiated into NSCs using PSC Neural Induction Medium as described in the manufacturer's protocol (Invitrogen). Briefly, pluripotent iPSC colonies were incubated in the Neural Induction Medium for 7 days and then transferred to Neural Expansion Medium. Differentiating NSCs were passaged every 4–6 days. At passage 5, the NCAM + NSCs were enriched using MACS with anti-PSA-NCAM microbeads (Miltenyi Biotec). At this stage, cells exhibit characteristic NSC morphologies and markers including Nestin, PAX6, SOX1, and SOX2. In passage 7, NSC was ready for subsequent RNA extraction.

**Induced neuronal (iN) cell differentiation**. iPSC-derived excitatory neurons were established using lentivirally introduced ectopic expression of Neurogenin 2 (*NGN2*)[81] with some modifications. HEK293T cells were kindly given by Professor Vijaya Ramesh's lab in the Center for Genomic Medicine at Massachusetts General Hospital. Lentiviruses were made in the HEK293T cells by co-transfection with VSV-G envelope expressing plasmid (pMD2.G addgene #12259), packaging plasmid (pCMV-dR8.2 dvpr #8455), and lentiviral transfer vectors (FUW-M2rtTA addgene #20342 and pTet-O-Ngn2-puro addgene #52047) using Lipofectamine 3000 reagents. Lentiviruses were harvested with the medium 48 h after transfection, pelleted by centrifugation (1,500 × g for 45 min) with Lenti-X Concentrator (Clontech), resuspended in DPBS, aliquoted, snap-frozen in liquid nitrogen, and stored in −80 °C. Lentiviral titer was determined using Lenti-X qRT-PCR Titration Kit (Clontech). On day -1, iPSCs from a control donor (GM8330)[82] were dissociated and plated as single cells in the medium with 10 μM Y-27632 dihydrochloride monohydrate (Santa Cruz Biotechnology). One hour after cell plating, iPSCs were transduced with lentiviruses carrying *NGN2* and M2rtTA overnight. For the transgene expression, on day 0 the culture medium was replaced with Neural maintenance medium[83] and doxycycline (2 mg/l, Clontech) was added into iPSC culture and gradually turned off from day 10. On day 1, the cells were selected with puromycin (1 ml/l, Gibco) for 48–72 h. On day 3, iN cells were dissociated with accutase and plated onto Matrigel-coated 12-well plates (2 × 10^5 cells/well) in Neural maintenance medium containing doxycycline (2 mg/l), human BDNF (10 μg/l, PeproTech), human NT-3 (10 μg/l, PeproTech); Ara-C (2 μM, Sigma) was added to the medium to inhibit astrocyte proliferation. From day 6, 50% of the medium in each well was exchanged every 3 days, preventing iN exposure to air. With each media change, Neural maintenance media was supplemented with BDNF (10 μg/l, PeproTech), human NT-3 (10 μg/l, PeproTech), and doxycycline (2 mg/l Clontech). The iNs were mature and ready for subsequent RNA extraction on day 24.

**Comparisons of patient-derived cerebral organoids with isogenic 16p11.2 lines**. We observed that 9500 genes were expressed in both the neural stem cells with 16p11.2 duplications and patient-derived cerebral organoids (SetA), out of which, 113 of these genes (1.2%) were differentially expressed with FDR ≤ 0.05 in both the neural stem cells and organoids, after excluding the genes in the 16p11.2 locus. In contrast, we observed that 9531 genes were expressed in both the induced neurons with 16p11.2 duplications and patient-derived cerebral organoids, out of which, 11 of these genes (0.12%) were differentially expressed with FDR ≤ 0.05 in both the induced neurons and organoids (OR = 10.4, 95% CI=[5.6, 21.5], Fisher's Exact Test *P* < 2.2 × 10^−16).

We used a more stringent criteria and observed that 34/9504 (0.36%) of the genes in both organoids and neural stem cells were similarly differentially expressed with fold changes of greater or <1 in the cases compared to the controls. In contrast, none of the 9526 genes that were expressed in both the induced neurons with 16p11.2 deletions and patient-derived cerebral organoids, were differentially expressed with FDR ≤ 0.05 in both neurons and organoids.

For the duplication lines, using a more stringent criteria, we observed that 63/ 9500 genes (0.66%) in both the organoids and neural stem cells were reciprocally differentially expressed with fold changes of >1 in the case organoids compared to control organoids, and fold changes of <1 in the neural stem cells with 16p11.2 duplications compared to wildtype, and vice versa. In contrast, 3/9,531 genes (0.031%) reciprocally differentially expressed with fold changes of >1 in the case organoids compared to control organoids, and fold changes of <1 in the

induced neurons with 16p11.2 duplications compared to wildtype, as well as vice versa (OR = 21.2, 95% CI=[6.9, 105.7], Fisher's Exact Test *P* = 5.7 ≤ 10^−16).

**Analysis of post-mortem brain samples with 15q11-13 duplications**. The differential expression results on post-mortem brain samples from the cortex with or without 15q11–13 duplications were downloaded from a prior publication[39]. We calculated CellScores for each of the 10 cell type clusters by using the *P*-values from the differential expression results for each gene. To calculate *P*(CellScore), we compared the observed CellScore from the post-mortem brain samples against the null CellScore distributions for each of the 10 cell type clusters generated by the permutations using the expression data from the cerebral organoids with 15q11–13 duplications, accounting for the precise numbers of genes used in the calculations of CellScores from the post-mortem brain samples. We calculated a weighted average *P*-value for the results from the cerebral organoids with 15q11–13 duplications and the results from the post-mortem brain samples with 15q11–13 duplications (Supplementary Data 9), which allows us to evaluate the combined CellScore results from the cerebral organoids and the post-mortem brain samples.

$$\text{Average}P(\text{CellScore}) = \frac{\left|\text{CellScore}_{\text{organoid}}\right|}{\left|\text{CellScore}_{\text{organoid}}\right| + \left|\text{CellScore}_{\text{postmortem}}\right|}$$
$$\times [-\log_{10}[P(\text{CellScore}_{\text{organoid}})]] + \frac{\left|\text{CellScore}_{\text{postmortem}}\right|}{\left|\text{CellScore}_{\text{organoid}}\right| + \left|\text{CellScore}_{\text{postmortem}}\right|} \quad (3)$$
$$\times [-\log_{10}[P(\text{CellScore}_{\text{postmortem}})]]$$

where $|\text{CellScore}_{\text{organoid}}|$ and $|\text{CellScore}_{\text{postmortem}}|$ are the absolute CellScore values calculated from the cerebral organoids and post-mortem brain samples respectively, and $P(\text{CellScore}_{\text{organoid}})$ and $P(\text{CellScore}_{\text{postmortem}})$ are the *P*(CellScore) values calculated from the cerebral organoids and post-mortem brain samples respectively.

**Identifying a minimum number of unique clusters from the Tanaka study**. The Tanaka study identified 24 cell clusters[19], and we calculated the pairwise percentage overlaps between genes from each pair of cell clusters (Supplementary Data 9). For each annotated cell cluster, we identified a minimal set of cell clusters with high percentage overlaps (≥50%), and highlighted the minimal set of cell clusters in blue in Supplementary Data 9.

**Calculation of overlaps and correlations with neurodevelopmental maps**. We used the cell type clusters from two large-scale scRNA-seq studies on brain organoids and fetal brains as neurodevelopmental maps[13,20], and calculated the percentage overlaps between the genes found in each cell type cluster from the Quadrato and Tanaka studies[1,19], and each cell type cluster from both neurodevelopmental maps (Supplementary Data 10). Using these percentage overlaps, we calculated the Pearson's correlations between each cell type cluster from both the Quadrato and Tanaka studies[1,19] (Supplementary Data 11).

**Calculation of GeneScore and *P*(GeneScore)**. There were 22 genes in the 16p11.2 locus that are expressed in the cerebral organoids, but two of the genes (*SULT1A3* and *QPRT*) were not found in the BrainSpan expression dataset, and were excluded from our candidate driver gene analyses. We calculated GeneScore for each gene *x* in a CNV locus using the total sum of the Pearson's correlation ($r^2_{x,y}$) of gene *x* with each gene y in the BrainSpan Project[84], multiplied by the −log₁₀-transformed *P*-values from the organoid differential expression results for gene *y* ($P_y$), and divided the scores by the total number of genes ($\text{Num}_y$) from the BrainSpan Project with correlations available for gene *x*.

We obtained a null distribution for GeneScore by performing 100,000 permutations (Supplementary Fig. 9), and performed linear regressions on the expression data for each permutation. Next, we calculated GeneScore for each gene *x* based on the permuted linear regression results. Since our observation and each permutation comprises of different combinations of individuals who have been assigned as pseudo-cases or pseudo-controls, we calculated a representative statistic ($\lambda$)[53], which is the ratio of the observed median to the expected median test statistic, to evaluate the *P*-value distribution in each permutation, and normalized the observed and permuted GeneScores with the inverse of $\log_{10}\lambda$:

$$\text{GeneScore}(x) = \frac{1}{\log_{10}\lambda} \sum_{\text{all} y} \frac{-\log_{10} P_y \times r^2_{x,y}}{\text{Num}_y} \quad (4)$$

We estimated the probability of the observed GeneScore for each gene *x* by comparing the observed GeneScore with the null distribution (GeneScore_permuted):

$$P(\text{GeneScore}(x)) = P(\text{GeneScore}_{\text{permuted}}(x) \geq \text{GeneScore}(x)) \quad (5)$$

To evaluate the cell type-specific GeneScores, we used the differential expression results from the same 100,000 permutations and calculated cell type-specific GeneScore using only the specific and non-specific genes in each cell cluster (c1-c10). To estimate the FDR for the cell type-specific GeneScores in the 16p11.2 locus, we sorted all the *P*-values calculated for the GeneScores from all clusters for

each locus, to obtain the distributions of *P*-values. For each locus, we used the 5[th] percentile *P*-value as the FDR threshold of 0.05, and 10th percentile *P*-value as the FDR threshold of 0.1.

**Gene ontology analyses**. Gene ontology analyses were performed using the online tool provided by Panther v17.0[85], available at: http://geneontology.org/.

**Reporting summary**. Further information on research design is available in the Nature Research Reporting Summary linked to this article.

## Data availability
The raw fastq data from RNA sequencing and whole-exome DNA sequencing generated in this study have been deposited in the SRA database under accession code PRJNA824347. The processed RNA sequence data generated in this study are available in tthe GEO database under accession code GSE200851.

## Code availability
All codes used in this study are available online at https://gitlab.com/elimlab/orgo-seq.

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

## Acknowledgements

We are grateful to all of the families at the participating Simons Simplex Collection (SSC) sites, as well as the principal investigators (A. Beaudet, R. Bernier, J. Constantino, E. Cook, E. Fombonne, D. Geschwind, R. Goin-Kochel, E. Hanson, D. Grice, A. Klin, D. Ledbetter, C. Lord, C. Martin, D. Martin, R. Maxim, J. Miles, O. Ousley, K. Pelphrey, B. Peterson, J. Piggot, C. Saulnier, M. State, W. Stone, J. Sutcliffe, C. Walsh, Z. Warren, E. Wijsman). This study was supported by the National Institutes of Health grants (NHGRI RM1HG008525 to G.M.C.; NIMH R01MH113279 to G.M.C. and B.A.Y.; NIA U01AG061835 to G.M.C., E.T.L., and Y.C.; NINDS R01NS093200 to J.F.G.; NICHD R01HD096362 to M.E.T.), the Simons Foundation for Autism Research (SFARI #573206 to M.E.T.), and Robert Wood Johnson Foundation grant (74178 to G.M.C.). C.A.W. is an Investigator of the Howard Hughes Medical Institute. Computing support was provided by the Harvard Medical School's Orchestra High-Performance Computing Group, which is partially supported by NIH grant NCRR 1S10RR028832-01.

## Author contributions

E.T.L., Y.C., and G.M.C. conceived the study. E.T.L. and Y.C. performed the cerebral organoid differentiation. P.D. implemented the Orgo-Seq package and wrote documentation on using Orgo-Seq. E.T.L., S.E., D.J.C.T., J.F.G., and M.E.T. developed the analyses for comparing the 16p11.2 patient and isogenic RNA sequence data. E.T.L., Y.C., and J.M.R. performed DNA extractions and library preparations. E.T.L., Y.C., Y.K.C., J.J.C., and K.M. performed RNA extractions. X.Z. performed the cryosectioning and immunostaining. E.T.L., Y.C., S.R., J.N.H., and G.M.C. developed the statistical methods and analyses. S.L. and M.J.B. performed the DNA and RNA sequence analyses. E.T.L., Y.C., C.A.W., B.A.Y., J.F.G., M.E.T., and G.M.C. acquired the samples and funding. E.T.L. and Y.C. wrote the manuscript with input from all authors.

## Competing interests

The authors declare no competing interests.
