## [Peer Review File · Nature Communications]

Orgo-Seq integrates single-cell and bulk transcriptomic data to identify cell type specific-driver genes associated with autism spectrum disorderEditorial Note: This manuscript has been previously reviewed at another journal that is not operating a transparent peer review scheme. This document only contains reviewer comments and rebuttal letters for versions considered at *Nature Communications*.

Reviewers' comments:

Reviewer #2 (Remarks to the Author):

To the authors, I wish you and your families health, happiness, and some semblance of stability during these trying times.

In their manuscript now titled "Data integration of bulk and single-cell transcriptomics from cerebral organoids and post-mortem brains to identify cell types and cell type specific driver genes in autism" Lim et al identify the cell type(s) and driver gene(s) in 16p11.2 deletion and 15q11-13 duplications using patient-derived human brain organoids. They conducted RNAseq on 1,420 organoids from 25 individuals, and developed "Orgo-Seq" to systematically quantify and identify the inherent variability in whole-transcriptome bulk RNA sequence data derived from the organoids. Their primary findings are the identification of neuroepithelial cells as a critical cell type for 16p11.2 deletions, the prioritization of KCTD13 as a cell type specific driver gene, and the validation of KCTD13 using CRISPR/Cas9-edited KCTD13 mosaic organoids. Since the original revision, the authors have clarified their methods and conclusions, and done a great job addressing concerns from my previous review. I am happy to now recommend this publication for publication in *Nature Communications*.

Reviewer #3 (Remarks to the Author):

This manuscript is a revision of a manuscript previously submitted to *Nature Genetics*. The paper analyzes two genomic copy number variations (CNVs), 16p11.2 deletion and 15q11-13 duplication, for their role in autism spectrum disorders (ASD). Using cerebral organoids grown from iPSCs of control donors and patients carrying said CNVs, the authors try to identify disease driving genes within the affected regions and find cell types driving the disease. The paper focusses on a candidate gene KCTD13 found in the 16p11.2 deletion, which the authors verify with a pooled knockout experiment. In the current version of the manuscript, the authors have moved the focus towards their data analysis platform Orgo-Seq that allows to identify both driver genes and cell types causative for ASD based on bulk RNAseq from patient-derived cerebral organoids when compared to published reference scRNAseq and post-mortem brain control datasets.

The authors address an interesting problem of high relevance to the field, namely that driving genes and cells are not easily identified in ASD patients with CNVs. Cerebral organoids grown from patient iPSCs are a potential surrogate for patient phenotypes. Thus, their phenotype needs to be compared to control data and the authors' Orgo-Seq pipeline is suggested to provide a platform to solve this problem.

Unfortunately, though, the paper has only been slightly changed and my initial concerns have been barely addressed. Clearly, one strength of the paper is the large amount of bulk RNAseq generated from cerebral organoids grown from multiple donor iPSCs. Unfortunately, however, the analysis of those data by Orgo-Seq is not really convincing. I feel there is still insufficient support to unequivocally demonstrate that Orgo-Seq allows for identification of ASD driver genes and cell types in patients with CNVs by way of analyzing cerebral organoids grown from their iPSCs. Below, I will point out some concerns about serious logical weaknesses that remain.

Major points:

1. The new focus of the paper is on the Orgo-Seq data integration framework. The major claim is that Orgo-Seq can identify driver genes and cell types causal for ASD based on cerebral organoids grown from patient iPSCs. To support this claim, the authors should reanalyze organoids on an individual donor basis. The assumption would be that one can identify genes and cell types based on their references.
2. To bolster their reference input wildtype basis, I want to once again point to state-of-the-art scRNAseq reference data from the Arlotta, Kriegstein and Treutlein labs among others. It may well be that these reference datasets improve the Orgo-seq basis to a point where better phenotyping may be possible.
3. To verify their driver gene identification, the authors perform a chimeric organoid experiment knocking out KCTD13. The authors provide FACS plots for their markers TRA1-60, Nestin and NeuN but unfortunately, only 3.88, 11.3 and 4.46% of cells are positive for any of the markers used. In my view, this raises serious concerns about the method used. Even assuming each cell is only positive for one of the markers, what are the remaining 80% of cells? How can the authors conclude that Nestin-positive neural progenitor cells are driving the phenotype if they only make up 11.3% of cells in their organoids?
4. The authors claim KCTD13 to be the gene causative for ASD in the 16p11.2 deletion. This is primarily based on published data and they cite very recent papers. They also performed the verification experiment targeting KCTD (see point 3). However, based on a revised Orgo-Seq analysis they also prioritize two additional genes, YPEL3 and INO80E, in the 16p11.2 deletion. Do these genes also hold up to their verification assay? As it stands, the authors turn their argument on their head with published data verifying their chimeric organoid model while Orgo-Seq predicts additional genes.

Reviewer #4 (Remarks to the Author):

While the authors have made some changes to the paper, much still remains to be done before this paper would be suitable for publication at this journal.

In general, the flow and framing of the paper are still very much unclear. There are many tangential topics in the text making the paper hard to understand. It is unclear why some of the analyses were performed (especially regarding the DE of the different 16p11 groups, but not limited to this). Additionally, while the authors state that they highlight the development of a framework most of the paper is not about this development and it still remains unclear

The following major changes are still required (organized by numbers of original critiques).

1. The comparison of CellScore to CIBERSORTx seems insufficient to prove this is a useful novel method. While CIBERSORTx is a deconvolution method CellScore doesn't aim to deconvolute the data (as the authors state). Rather, CellScore is a method to test for enrichment of cell types using cell-specific genes and would therefore benefit from comparisons to similar enrichment tests. If the paper is a methods development paper there needs to be some conceptual schema of why this method works better than existing methods in a main figure.
2. The figures are still not at all sufficient. The authors state they have checked the figures but there are still figures lacking y-axis labels. The legends need to be re-written as well to much more detailed results/statistics. Many supplementary figures should be included as panels in the main figures - there

is very little substance in the main figures. It is very hard to understand what the results are by looking at the figures.

3. It is still unclear why the authors used different thresholds for the most variable genes. The fact they yield the same number of genes is irrelevant and should not be a consideration as this will change from experiment to experiment. Rather the authors should use statistical justification for a cutoff and use it in both instances. If the genes are variable because they show low expression, use a RPKM cutoff instead of a variability cutoff. If the genes are highly expressed and variable, they should be included in the analysis.

4. While the authors state that they have modified the text to clarify the differential expression much of the text presented as new, it is essential the same text from the original submission and the authors did not respond to our comment. - is there any overlap between the different sets (here we only see SetA compared with the 15q11-13 deletion). Are the logFC similar?

5. In this paper, organoids were grown for 46 days, while the cell types derived from single cell RNAseq were ascertained from 3 and 6 month-old organoids. The authors claim that their method is tolerant to these time differences, but in order to make this claim they need to show much more data, as this comprises very large differences in developmental stage and cell composition.

a. The authors do not show any evidence that the cell types that they are using in the analysis are even found at this early stage. For example - cluster c4 "cells with forebrain markers" - are these cells/markers present in the organoid data?

b. One alternative hypothesis is that a difference in the cell specific markers could mean a developmental shift in the treated vs. control and not a cell enrichment shift. Can the authors show data for the cell specific markers across development to rule out this hypothesis or at least discuss this in interpreting their data?

6. The authors did not address the misclassification and while they claim that it was validated in another data set (from the same lab) there are many published single cell data sets and the authors should really validate their methods using many of these data sets.

7. (Additional comment). For the odds ratio statistics comparing the isogenic with the patient-derived lines, is this statistic (page 9 of the response to reviewers) comparing the overlap in DEGs between the neural stem cells vs. the iNeurons? Or is this showing that the neural stem cell/patient derived cerebral organoid overlap is more significant than would be expected by chance? Make it clear what statistical comparisons are being done in the text and figure legends.

Minor issues not addressed (from critiques):

10. Using all samples, even those from the same individual, as independent samples in the linear model falsely inflates the number of DEG (see Germain and Testa 2017 Stem Cell Reports). Instead, a mixed linear model should be used to account for multiple samples coming from the same individual. - this was not addressed. The assumption of independence even if it was used to increase the number of permutations is wrong.

15. There are still more sections in the results that need to be moved to the discussion.

We thank all the reviewers for taking the time to provide detailed and insightful comments that have greatly improved our revised manuscript. We summarize a few key revisions to our manuscript, and provide a detailed point-by-point response below.

Emphasizing the focus of the Orgo-Seq framework

We apologize that our previous manuscript did not sufficiently highlight the novelty and focus of the Orgo-Seq framework. We have now rewritten our manuscript and the new title of our manuscript is “Multi-transcriptomics data integration of cerebral organoids and post-mortem brains to identify cell type specific co-expressed driver genes in autism”.

Briefly, we emphasized that the goal of Orgo-Seq is not simply to perform cell type deconvolution or enrichment (which can be performed using existing tools such as CIBERSORT, CIBERSORTx and xCell). Instead, Orgo-Seq is a framework to enable the discovery of cell type specific co-expression (such as cell type specific co-expressed driver genes). By performing data integration of single-cell RNA sequence (scRNA-seq) and bulk RNA sequence (bRNA-seq) data, we can leverage on the strengths of performing unbiased critical cell type discovery from scRNA-seq data, but also overcome weaknesses in current scRNA-seq technologies for the discovery of cell type specific co-expressed genes. We had also clarified that in order to achieve the identification of critical cell type co-expressed driver genes, we took a 2-step approach with Orgo-Seq: to first identify the critical cell types (*CellScore*), and next to identify cell type specific driver genes (*GeneScore*). To improve readability of our manuscript, we have removed several tangential parts of the manuscript and cited prior literature instead, or moved sections into our supplementary methods.

Integration of additional large-scale single-cell RNA seq data

As Reviewers 3 and 4 had pointed out, there are several state-of-the-art, large-scale single-cell RNA sequence (scRNA-seq) datasets that had been generated and published from several groups. There is a recent paper by Tanaka Y *et al.* (Cell Reports 2020) that aggregated all the scRNA-seq datasets from 8 different brain organoid protocols and fetal brains that were previously published by multiple groups, comprising of ~190,000 single cells. We used their aggregated scRNA-seq dataset to identify 11 unique cell type clusters, and have now included analyses on this larger-scale scRNA-seq using the Orgo-Seq framework. Our previous results were based on the scRNA-seq dataset by Quadrato G *et al.* (Nature 2017) comprising of ~67,000 single cells, so the Tanaka dataset has ~3 times more single cells.

Replication of critical cell types using the Tanaka scRNA-seq data

Previously by using the Orgo-Seq framework, we identified that the c9 (neuroepithelial cell) cluster and c6 (unknown) cluster from the Quadrato dataset were critical cell types for the 16p11.2 locus. In our updated analyses using the Orgo-Seq framework with the Tanaka dataset, we found that there was only 1 critical cell type cluster (CC3 comprising of cortical excitatory neurons) for the 16p11.2 locus. We also found that there were strong correlations between the c9 cluster from the Quadrato study and the CC3 cluster from the Tanaka study.

Fine-mapping of cell type cluster identities using reconstituted neurodevelopmental maps

Reviewer 4 had raised concerns about potential inconsistencies in cell type cluster identification from different groups/studies generating scRNA-seq data from brain organoids. Velasco S *et al.* (Nature 2019) and Eze UC *et al.* (Nature Neuroscience 2021) had generated detailed neurodevelopmental maps reconstituted from large-scale scRNA-seq data from brain organoids and fetal brains. We have now performed systematic comparisons between cell type clusters identified from different studies (the Quadrato and Tanaka studies), with the cell type clusters in

both neurodevelopmental maps, and found that both the c9 cluster and CC3 cluster map to the same critical cell types implicating immature neurons and intermediate progenitor cells for the 16p11.2 locus.

Replication of candidate driver genes using the Tanaka scRNA-seq data

We used the Orgo-Seq framework with the Tanaka dataset to identify candidate driver genes using the CC3 cluster, and identified *YPEL3* with $FDR \leq 0.05$, and both *KCTD13* and *INO80E* with $FDR \leq 0.1$. We had also identified *CDIPT* with $FDR \leq 0.1$. Thus, across both datasets and analyses, we replicated our results that *YPEL3*, *KCTD13* and *INO80E* are likely to be bona fide candidate driver genes impacting immature neurons and intermediate progenitor cells for the 16p11.2 locus.

Our point-by-point responses (in blue text) to the reviewers' comments (in black text) is as follow. References to updated text in the revised manuscript are highlighted in red text.

Point-by-point response

Reviewer #2 (Remarks to the Author):

To the authors, I wish you and your families health, happiness, and some semblance of stability during these trying times. In their manuscript now titled "Data integration of bulk and single-cell transcriptomics from cerebral organoids and post-mortem brains to identify cell types and cell type specific driver genes in autism" Lim et al identify the cell type(s) and driver gene(s) in 16p11.2 deletion and 15q11-13 duplications using patient-derived human brain organoids. The conducted RNAseq on 1,420 organoids from 25 individuals, and developed "Orgo-Seq" to systematically quantify and identify the inherent variability in whole-transcriptome bulk RNA sequence data derived from the organoids. Their primary findings are the identification of neuroepithelial cells as a critical cell type for 16p11.2 deletions, the prioritization of *KCTD13* as a cell type specific driver gene, and the validation of *KCTD13* using CRISPR/Cas9-edited *KCTD13* mosaic organoids. Since the original revision, the authors have clarified their methods and conclusions, and done a great job addressing concerns from my previous review. I am happy to now recommend this publication for publication in Nature Communications.

We sincerely thank the reviewer for her/his positive comments and well wishes, and we are very grateful to all reviewers for their very fast and thorough reviews, especially during these times.

Reviewer #3 (Remarks to the Author):

This manuscript is a revision of a manuscript previously submitted to Nature Genetics. The paper analyzes two genomic copy number variations (CNVs), 16p11.2 deletion and 15q11-13 duplication, for their role in autism spectrum disorders (ASD). Using cerebral organoids grown from iPSCs of control donors and patients carrying said CNVs, the authors try to identify disease driving genes within the affected regions and find cell types driving the disease. The paper focusses on a candidate gene *KCTD13* found in the 16p11.2 deletion, which the authors verify with a pooled knockout experiment. In the current version of the manuscript, the authors have moved the focus towards their data analysis platform Orgo-Seq that allows to identify both driver genes and cell types causative for ASD based on bulk RNAseq from patient-derived cerebral organoids when compared to published reference scRNAseq and post-mortem brain control datasets.

The authors address an interesting problem of high relevance to the field, namely that driving genes and cells are not easily identified in ASD patients with CNVs. Cerebral organoids grown from patient iPSCs are a potential surrogate for patient phenotypes. Thus, their phenotype

needs to be compared to control data and the authors' Orgo-Seq pipeline is suggested to provide a platform to solve this problem.

We thank the reviewer for highlighting the importance of the biological questions that we are addressing with our current manuscript, which were not easily addressable by using blood or brain samples from ASD patients, or by using animal models.

Unfortunately, though, the paper has only been slightly changed and my initial concerns have been barely addressed. Clearly, one strength of the paper is the large amount of bulk RNAseq generated from cerebral organoids grown from multiple donor iPSCs. Unfortunately, however, the analysis of those data by Orgo-Seq is not really convincing. I feel there is still insufficient support to unequivocally demonstrate that Orgo-Seq allows for identification of ASD driver genes and cell types in patients with CNVs by way of analyzing cerebral organoids grown from their iPSCs. Below, I will point out some concerns about serious logical weaknesses that remain.

Reviewer 3 Major points:

1. The new focus of the paper is on the Orgo-Seq data integration framework. The major claim is that Orgo-Seq can identify driver genes and cell types causal for ASD based on cerebral organoids grown from patient iPSCs. To support this claim, the authors should reanalyze organoids on an individual donor basis. The assumption would be that one can identify genes and cell types based on their references.

The reviewer has raised an interesting new angle about analyzing the organoids on an individual donor basis, and we agree that the development of new methods for analyzing cerebral organoids from individual donors for identifying cell types and driver genes in an individual patient, is of great significance for diagnosing disease subtypes or personalized genetic causes of the disease (precision medicine).

However, the reviewer is asking a different question from what our current manuscript and Orgo-Seq aims to address. Orgo-Seq compares donor-derived cerebral organoids from many individuals with the same genetic risk loci, to enable the discovery of novel critical cell type(s) and cell type specific driver genes. The development of methods for personalized diagnostics will require previously identified critical cell type(s) and cell type specific driver genes.

We are really excited about the reviewer's question if we can eventually achieve personalized diagnoses of individuals using cerebral organoids, and have also added the following text to conclude our discussion:

"As a future direction, it will be exciting to explore the possibility of developing a precision medicine framework to rapidly identify critical cell types and cell type specific driver genes in individual donors, and the framework can complement DNA sequencing to enable the identification of putative causal cell types and cell type specific genes and gene networks in an individual patient for personalized diagnostics."

Reviewer 3 Major points:

2. To bolster their reference input wildtype basis, I want to once again point to state-of-the-art scRNAseq reference data from the Arlotta, Kriegstein and Treutlein labs among others. It may well be that these reference datasets improve the Orgo-seq basis to a point where better phenotyping may be possible.

We thank the reviewer for pointing out these additional high-quality datasets. We found that Yoshiaki Tanaka and In-Hyun Park had harmonized and projected a total of 190,022 single cells post-quality control, across 8 different scRNA-seq datasets generated from brain organoids and fetal brains (including the datasets mentioned by the reviewer). Our updated manuscript now includes analyses using the Tanaka harmonized scRNA-seq data, and as mentioned in our summary above, our results for the critical cell types and candidate driver genes in the 16p11.2 locus have been replicated, and we believe that our manuscript is much stronger now because of the reviewer's suggestion.

The new sections in our updated manuscript are as follow:

Introduction

"We applied Orgo-Seq for two ASD-associated copy number variants (CNVs) in the 16p11.2 and 15q11-13 loci¹⁶⁻¹⁸, by integrating 3 sets of transcriptomics datasets: bRNA-seq data that we generated from donor-derived cerebral organoids, previously published scRNA-seq data from cerebral organoids and fetal brains^{1,13,19,20}, and previously published bRNA-seq data from human post-mortem brain samples in the BrainSpan Project²¹. Using an initial scRNA-seq dataset from 66,889 single cells¹, we initially observed that neuroepithelial cells are perturbed in donor-derived cerebral organoids from individuals with deletions in 16p11.2 compared to individuals without the deletions, and that 3 of the genes in the locus (*YPEL3*, *KCTD13* and *INO80E*) are likely to be candidate driver genes functioning in neuroepithelial cells. Using a larger scRNA-seq dataset comprising of 190,022 cells¹⁹ from brain organoids differentiated using 8 different protocols and fetal brains^{1,7-13,22}, and two neurodevelopmental maps constructed from scRNA-seq on brain organoids and fetal brains to fine-map the critical cell types^{13,20}, we replicated the critical cell type that was initially discovered, and were able to pinpoint the identity of the critical cell type more precisely during neurodevelopment to immature neurons and intermediate progenitor cells for the 16p11.2 locus. We also replicated our initial results that *YPEL3*, *KCTD13* and *INO80E* are cell type specific candidate driver genes."

Results

"Fine-mapping of cell type identities using large-scale neurodevelopmental maps point to the role of immature neurons and intermediate progenitor cells for the 16p11.2 locus.

A recent study by Tanaka Y *et al.* had re-analyzed 190,022 cells from brain organoids differentiated using 8 different protocols and fetal brains^{1,7-13,22}, and identified 24 cell type clusters¹⁹. We systematically compared the percentage overlaps among genes across the 24 cell type clusters to identify 11 unique clusters (CC1-11; Supplementary Table 13). We calculated *CellScores* using the 11 cell type clusters for 16p11.2, and found that there was only the cell type cluster comprising of cortical excitatory neurons (CC3) that had an $FWER \leq 0.05$ ($P(\text{CellScore}) = 3.6 \times 10^{-4}$, Fig. 3C). Interestingly, we did not observe any association for 16p11.2 with the neuroepithelial cell cluster (CC7) in the Tanaka study ($P(\text{CellScore}) = 0.11$).

To ensure consistencies in assigning cell type identities across different studies, and to fine-map the critical cell types more precisely during neurodevelopment, we used two neurodevelopmental maps that were reconstituted from scRNA-seq data on brain organoids and fetal brains. The first neurodevelopmental map by Velasco S *et al.* comprised of 12 cell types¹³, and we calculated the percentage overlap among genes from each cell type cluster reported by Quadrato G *et al.* and Tanaka Y *et al.*^{1,19} (Supplementary Table 14). We found that the neuroepithelial cell cluster (c9) from the Quadrato study overlapped most closely with immature projection neurons (mean overlap = 0.37%), and that the unknown cell type (c6) from the Quadrato study overlapped most closely with outer radial glia cells (mean overlap = 0.29%). The CC3 cluster from the Tanaka study overlapped most closely with immature projection neurons in the neurodevelopmental map (mean overlap = 1.14%), similar to the c9 cluster from the Quadrato study.

We used a second neurodevelopmental map by Eze UC *et al.* that comprised of 6 cell types²⁰, and calculated the percentage overlap among the genes from each cell type cluster reported by the Quadrato and Tanaka studies (Supplementary Table 14). We found that the c9 cluster from the Quadrato study overlapped most closely with the neuronal and intermediate progenitor cell clusters (mean overlaps = 8.1% and 5.4% respectively). Similarly, the CC3 cluster from the Tanaka study overlapped most closely with the neuronal and intermediate progenitor cell clusters (mean overlaps = 19.8% and 11.1% respectively). These results suggest that the critical cell types for the 16p11.2 locus are likely to be immature neurons and intermediate progenitor cells.

To evaluate the degree of independence among the genes in the c9 cluster from the Quadrato study and the CC3 cluster from the Tanaka study, we calculated the correlations between the mean overlaps across the two neurodevelopmental maps for both the c9 and CC3 clusters (Supplementary Table 15). We observed high correlations between both the c9 and CC3 clusters using both neurodevelopmental maps ($r=0.71$, $P=9.1\times 10^{-3}$; $r=0.95$, $P=3.6\times 10^{-3}$). However, there were stronger correlations between the CC3 and c5 clusters ($r=0.96$, $P=8.7\times 10^{-7}$; $r=0.98$, $P=6.3\times 10^{-4}$), even though the c5 cluster was not implicated as the critical cell type from the 16p11.2 donor-derived organoids. This indicates that there is likely to be independence among the genes implicating the c9 and CC3 clusters as critical cell types in the 16p11.2 locus.

Replication of driver gene results for the 16p11.2 locus by integrating a large-scale scRNA-seq dataset

To replicate our results for the 16p11.2 locus, we calculated *GeneScores* using the cell type specific genes in the CC3 cluster from the Tanaka study. *YPEL3* was similarly prioritized as a high-confidence candidate driver gene at $FDR\leq 0.05$, and both *KCTD13* and *INO80E* were prioritized at $FDR\leq 0.1$ (Fig. 3F, Supplementary Tables 18-19). Another gene (*CDIPT*) was also prioritized at $FDR\leq 0.1$.

Reviewer 3 Major points:

3. To verify their driver gene identification, the authors perform a chimeric organoid experiment knocking out *KCTD13*. The authors provide FACS plots for their markers TRA1-60, Nestin and NeuN but unfortunately, only 3.88, 11.3 and 4.46% of cells are positive for any of the markers used. In my view, this raises serious concerns about the method used. Even assuming each cell is only positive for one of the markers, what are the remaining 80% of cells?

Based on the scRNA-seq data from brain organoids that had been published, there are additional cell types within the cerebral organoids that will not be positive for TRA-1-60, Nestin or NeuN. For instance, glia cells such as microglia and mature astrocytes will not be positive for any of the 3 markers.

In addition, we had used a highly stringent approach to gating the cells during FACS sorting, resulting in smaller fractions of positive cells for TRA-1-60, Nestin or NeuN. If we use less stringent gating, there will be greater fractions of positive cells for the 3 markers.

We updated our main text to clarify that we had focused on 3 groups of well-validated cell types in the mosaic organoid experiments:

“We focused on a subset of three groups of well-validated cell types to validate our results from Orgo-Seq, and selected four antibody markers for FACS – NeuN for neuronal cells, Nestin for neural progenitor cells, TRA-1-60 for stem cells and mouse IgG2A as a negative control (Supplementary Fig. 10).”

Reviewer 3 Major points:

3. (cont). How can the authors conclude that Nestin-positive neural progenitor cells are driving the phenotype if they only make up 11.3% of cells in their organoids?

We will like to clarify that FACS sorting is a selection assay, and the mosaic organoid validation system that we used is adapted from a similar system that our group and other groups had reported using yeast and bacteria (FlowSeq by Kosuri S *et al.* PNAS 2013; Raveh-Sadka T *et al.*, Nature Genetics 2012; Sharon E *et al.*, Nature Biotech 2012). We have now named the method as oFlowSeq for “organoid FlowSeq”, and cited our previous publication describing the system (Kosuri S *et al.*).

We have also provided a schematic to provide more clarity to the oFlowSeq system in the supplementary methods section.

“Schematic of sequencing results from oFlowSeq using mosaic cerebral organoids

The figure on the left illustrates an example where we dissociate the CRISPR-edited organoids and FACS sort equal numbers of cells for 3 sets of genotypes: 1) control cells with no edits in the *KCTD13* gene, 2) cells with a deleterious mutation in *KCTD13* that impacts protein function (red triangles), and 3) cells with a benign mutation in *KCTD13* that does not impact protein function (blue triangles).

In this example, there are 11 control unedited cells used for sorting (3 TRA-1-60⁺ cells, 3 Nestin⁺ cells and 5 NeuN⁺ cells).

Similarly, 11 edited cells that are heterozygous for a deleterious mutation in *KCTD13* were used for sorting (2 TRA-1-60⁺ cells, 5 Nestin⁺ cells and 4 NeuN⁺ cells). If the deleterious mutation in *KCTD13* results in increased numbers of Nestin⁺ cells, there will be an increased proportion of Nestin⁺ cells with the deleterious mutation, compared to TRA-1-60⁺ or NeuN⁺ cells with the deleterious mutation.

Another 11 edited cells that are heterozygous for a benign mutation in *KCTD13* were used for sorting, and we expect the proportions of cell types with the benign mutation to be similar to the proportions of cell types for the control unedited cells.

In this example, after FACS sorting using TRA-1-60 as a marker, followed by targeted sequencing of the TRA-1-60⁻ cells, we expect to count 33 reads without any mutations in *KCTD13* (66% of all reads), 9 reads with the deleterious mutation in *KCTD13* (18% of all reads), and 8 reads with the benign mutation in *KCTD13* (16% of all reads). For the TRA-1-60⁺ cells, we expect to count 11 reads without any mutations in the gene (69%), 2 reads with the deleterious mutation (13%), and 3 reads with the benign mutation (19%).

Similarly, the expected numbers and percentages of reads for the Nestin⁻ versus Nestin⁺ cells, and NeuN⁻ versus NeuN⁺ cells, are illustrated in the example. If the deleterious mutation in *KCTD13* results in increased numbers of Nestin⁺ cells, then we expect the Nestin⁺ cells to harbor

a higher percentage of the deleterious mutation compared to the percentage of the mutation in Nestin⁻ cells.

Based on the results, we calculated an odds ratio for each mutation in each sorted cell population. If there are benign mutations in *KCTD13* that are not differentially represented in a cell type (such as Nestin⁺ cells), then the distribution of odds ratios for these benign mutations will center around a mean odds ratio of 1, illustrated by the blue distribution in the figure on the right. However, if some of the mutations in *KCTD13* are deleterious and do affect the proportions of cell types, then these deleterious mutations will show a distribution of odds ratios that center around a mean odds ratio of greater than 1, illustrated by the pink distribution in the figure on the right.”

Reviewer 3 Major points:

4. The authors claim *KCTD13* to be the gene causative for ASD in the 16p11.2 deletion. This is primarily based on published data and they cite very recent papers.

We will like to clarify that because we are developing a novel data integration framework, we are primarily using published data to benchmark and validate our results and framework. The papers that we cite are recent because relatively new technologies and tools, as well as new resources, have enabled these studies to be conducted. The generation of most of the donor-derived iPSCs with 16p11.2 deletions was only reported only recently by Theo Palmer’s group (Roth JG, eLife 2020). Some of the iPSCs in our work were also from an earlier publication by Lauren Weiss’ group (Deshpande A, Cell Reports 2017). We obtained the donor iPSCs from both recent studies, and our manuscript is the first (or among the first) to report results from 16p11.2 deletion donor-derived cerebral organoids.

Reviewer 3 Major points:

4. (cont) They also performed the verification experiment targeting *KCTD* (see point 3). However, based on a revised Orgo-Seq analysis they also prioritize two additional genes, *YPEL3* and *INO80E*, in the 16p11.2 deletion. Do these genes also hold up to their verification assay? As it stands, the authors turn their argument on their head with published data verifying their chimeric organoid model while Orgo-Seq predicts additional genes.

We will like to clarify that the version of our manuscript prior to the initial submission had already prioritized *YPEL3* and *INO80E* as driver genes (besides *KCTD13*) using the Orgo-Seq framework. In this current revised manuscript, we included more scRNA-seq data from additional publications, and have now replicated the results for all 3 genes (*YPEL3*, *INO80E* and *KCTD13*) as cell type specific driver genes in immature neurons and intermediate progenitor cells for the 16p11.2 locus in our new analyses.

The reviewer had raised a question that is of great interest in the field, and Reviewer 2 had similarly asked about this in the previous review. In the zebrafish study by Golzio C *et al.*, the authors found that *KCTD13* is the only driver gene in the 16p11.2 deletion locus modulating the proportions of neural progenitor cells (or head sizes) in zebrafish. While our Orgo-Seq results found that *KCTD13* is a driver gene in the locus using patient-derived cerebral organoids, we did not find that there was a clear transcriptomic co-expression signature pointing to only *KCTD13* and not the other genes in the 16p11.2 locus. Reviewer 2 had asked if *KCTD13* is sufficient and necessary for driving the 16p11.2 phenotype, i.e. if *KCTD13* is the only driver

gene in the 16p11.2 locus. Since the publication by Golzio C *et al.*, there have been supporting evidence for the role of other driver genes in the 16p11.2 locus, and our results similarly support evidence for the role of multiple driver genes in the 16p11.2 locus.

In our previous response letter to address Reviewer 2's question, we had also elaborated on additional supporting evidence from our work that do not support the result that *KCTD13* is the only driver gene in the locus (shown below in green text).

While we provide direct evidence for the role of *KCTD13* affecting the proportions of neuroepithelial cells, Orgo-Seq also prioritizes 2 other genes in the 16p11.2 locus with $FDR \leq 0.05$. These 2 genes are less well-studied and more research is needed in the future to understand the role of these genes in the locus, and the interactions of these genes with *KCTD13*.

Kizner V *et al.* had also reported a list of 13 cell cycle associated genes that were expressed at significantly lower levels in the *KCTD13*-deficient NPCs compared to wildtype. None of these 13 genes are expressed at significantly lower levels in the patient organoids with 16p11.2 deletions. This suggests that deletions in *KCTD13* in human NPCs is insufficient to recapitulate the transcriptomic perturbations identified in the patient organoids with 16p11.2 deletions. We have added a new section on these results supporting the role of multiple driver genes in the 16p11.2 locus.

The goal of comparing the Orgo-Seq results with published results on *KCTD13* is to validate our multi-transcriptomics data integration approach (Orgo-Seq), and not to validate the mosaic organoid experiments (oFlowSeq). We had optimized methods for the mosaic organoid experiments, but we did not develop new methods. Similar methods had been developed previously by our group and other groups using yeast and bacteria (Kosuri S *et al.*, PNAS 2013; Raveh-Sadka T *et al.*, Nature Genetics 2012; Sharon E *et al.*, Nature Biotech 2012).

We thank the reviewer for raising this question, which led us to realize that we should have cited the previous papers. Our group had published previously using a similar FACS-based method, which was termed "FlowSeq" (Kosuri S *et al.*, PNAS 2013). However, the application using FlowSeq was for bacteria instead of human cerebral organoids, and addressed a different set of questions. We adapted an approach similar to FlowSeq for application to human mosaic cerebral organoids, and to address a different set of questions in ASD genetics, so we did not use the term FlowSeq to refer to our mosaic organoid experiments. For clarity, we have now termed the mosaic cerebral organoid approach as oFlowSeq for "organoid FlowSeq".

We have now updated our manuscript to provide citations to the previous publications on FlowSeq and similar approaches:

Introduction

"Given that immature neurons are enriched for Nestin⁺ cells²³⁻²⁵, we adapted a previous framework (FlowSeq)²⁶ to create a mosaic cerebral organoid framework using CRISPR/Cas9 editing (termed oFlowSeq for "organoid FlowSeq") to validate one of our key findings from Orgo-Seq for the 16p11.2 locus, that *KCTD13* is one of the driver genes in the locus modulating the proportions of Nestin⁺ cells in cerebral organoids. Our work presents a quantitative framework to identify cell types and cell type specific driver genes in a complex disease by integrating bRNA-seq and scRNA-seq from donor-derived cerebral organoids and human post-mortem brains (Orgo-Seq) and a CRISPR/Cas9 based mosaic cerebral organoid system (oFlowSeq) to validate the findings from the donor-derived cerebral organoids."

Results

“To provide further validation that *KCTD13* is one of the driver genes in the 16p11.2 locus modulating the proportions of immature neurons in the patient-derived organoids, and to resolve prior conflicting results from *KCTD13*-deficient animal models³⁹⁻⁴¹, we used a CRISPR-based approach⁶¹⁻⁶⁴ to directly measure the effects of knockouts in cerebral organoids, and adapted a fluorescence activated cell sorting (FACS) based approach that our group and other groups had previously described using bacteria and yeast^{26,65,66}. We termed the adapted approach as oFlowSeq for “organoid FlowSeq”.”

Reviewer #4 (Remarks to the Author):

While the authors have made some changes to the paper, much still remains to be done before this paper would be suitable for publication at this journal.

In general, the flow and framing of the paper are still very much unclear. There are many tangential topics in the text making the paper hard to understand. It is unclear why some of the analyses were performed (especially regarding the DE of the different 16p11 groups, but not limited to this). Additionally, while the authors state that they highlight the development of a framework most of the paper is not about this development and it still remains unclear.

We have now substantially rewritten our manuscript to emphasize the goal of Orgo-Seq, and have removed much of the tangential topics in the text, as well as cited published literature for clarification.

Reviewer 4:

The following major changes are still required (organized by numbers of original critiques).

1. The comparison of CellScore to CIBERSORTx seems insufficient to prove this is a useful novel method. While CIBERSORTx is a deconvolution method CellScore doesn't aim to deconvolute the data (as the authors state). Rather, CellScore is a method to test for enrichment of cell types using cell-specific genes and would therefore benefit from comparisons to similar enrichment tests. If the paper is a methods development paper there needs to be some conceptual schema of why this method works better than existing methods in a main figure.

We apologize that the goal of Orgo-Seq was not articulated clearly previously, and we have rewritten the manuscript to clarify this. As explained in our summary, Orgo-Seq is not an approach to simply identify critical cell types. The reviewer correctly stated that there are several existing methods for doing so, such as CIBERSORTx, CIBERSORT and xCell. In the xCell paper (Aran D *et al.*, Genome Biology 2017), the authors had extensively compared a cell type enrichment approach (xCell) to a deconvolution approach (CIBERSORT). Both xCell and CIBERSORT used bRNA-seq data from homogeneous cell populations as a reference panel, and not scRNA-seq data. However, the comparisons between a cell type enrichment approach versus a deconvolution approach is beyond the scope and aim of our manuscript, so we referenced the xCell paper instead.

We will also like to further emphasize the novelty and motivation behind Orgo-Seq. Ideally, scRNA-seq can be performed directly on the donor-derived cerebral organoids to identify cell type specific co-expressed driver genes, i.e. which of the genes in the 16p11.2 locus are co-expressed in specific cell types associated with ASD. However, the limited capture efficiencies of current high-throughput scRNA-seq technologies do not allow for these analyses to be

performed using scRNA-seq data alone. So Orgo-Seq was developed to address this limitation, by integrating bulk and scRNA-seq data from brain organoids to identify cell type specific co-expression such as cell type specific driver genes. Orgo-Seq is not just conceptually different from deconvolution/enrichment methods, but also, Orgo-Seq is addressing a different issue that is not addressed by deconvolution/enrichment methods. The more accurate comparison is between the Orgo-Seq results and analyses with what we could achieve using scRNA-seq alone or bulk RNA sequencing alone.

We have now rewritten our manuscript to emphasize this, as well as updated Figure 1 in our main text to illustrate that Orgo-Seq is an integrative platform to combine the strengths of bRNA-seq and scRNA-seq for cell type specific co-expression discovery.

Figure 1: Orgo-Seq framework to identify cell type specific co-expressed driver genes.

(A) Figure illustrating the strengths and weaknesses of bRNA-seq and scRNA-seq, and what Orgo-Seq can achieve by integrating both types of datasets.

(B) A schematic of the Orgo-Seq framework to integrate bRNA-seq data from patient-derived brain organoids with scRNA-seq data from control brain organoids and bRNA-seq data from post-mortem human brains (BrainSpan), for the discovery of critical cell types and cell type specific driver genes.

“Data integration of bRNA-seq data from donor-derived cerebral organoids and scRNA-seq data from control organoids identifies critical cell types for 16p11.2 deletions and 15q11-13 duplications

Deletions in 16p11.2 are significantly associated with ASD but not with schizophrenia, whereas duplications in 16p11.2 are associated with both ASD and schizophrenia^{6,32,33}. Clinical studies have shown that individuals with 16p11.2 deletions have increased brain sizes, and individuals with duplications in the same locus have decreased brain sizes^{32,34,35}. Mouse models with 16p11.2 deletions or duplications similarly show an increase or reduction in brain sizes and in the proportions of neural progenitor cells³⁶⁻³⁸. A systematic perturbation of all genes in the 16p11.2 locus using head sizes as the phenotypic readout in zebrafish identified *KCTD13* as the only driver gene in the locus modulating the proportion of neural progenitor cells³⁹. However, recent studies in mice and zebrafish with deleted *KCTD13* did not observe increased brain sizes

or neurogenesis in these mutant animal models^{40,41}. In the absence of human fetal brains with 16p11.2 deletions that could be used to resolve these conflicting results from animal models⁴², the use of donor-derived cerebral organoids could be good models to provide supporting results.

To accomplish this, we would have to identify which cell type specific co-expressed gene(s) from the donor-derived cerebral organoids are misregulating the proportions of critical cell types in cases versus controls (Fig. 1B). We developed a two-step solution where we first identified the critical cell types that were disproportionately affected in cases versus controls using bRNA-seq data from the donor-derived cerebral organoids, and a second step where we identified which of the genes in the CNV loci were disproportionately misregulating cell type specific expression of genes outside the CNV loci between cases versus controls.

There are two general approaches to identifying critical cell types from bRNA-seq data: deconvolution methods such as CIBERSORT and CIBERSORTx, or cell type enrichment methods such as xCell^{43,44}. Previously, when using bRNA-seq data from pure cell types as a reference panel⁴³, it was shown that a cell type enrichment approach (xCell) outperforms a deconvolution approach (CIBERSORT). We sought to develop a cell type enrichment based approach for bRNA-seq data from cerebral organoids, by using scRNA-seq data from brain organoids and fetal brains as a reference panel.”

Reviewer 4 Major issues:

2. The figures are still not at all sufficient. The authors state they have checked the figures but there are still figures lacking y-axis labels. The legends need to be re-written as well to much more detailed results/statistics. Many supplementary figures should be included as panels in the main figures - there is very little substance in the main figures. It is very hard to understand what the results are by looking at the figures.

We apologize for the confusion and have checked all the figures to ensure that they all have y-axis labels. We have also shortened all the legends to remove detailed results/statistics. We have gone through all the supplementary figures and felt that while the supplementary figures were important for quality control, demonstrating the validity of the data or conceptualization of our methods, they were not directly contributing to visualizing the results of the main text (unlike our main figures).

Reviewer 4 Major issues:

3. It is still unclear why the authors used different thresholds for the most variable genes. The fact they yield the same number of genes is irrelevant and should not be a consideration as this will change from experiment to experiment. Rather the authors should use statistical justification for a cutoff and use it in both instances. If the genes are variable because they show low expression, use a RPKM cutoff instead of a variability cutoff. If the genes are highly expressed and variable, they should be included in the analysis.

We will like to clarify that we did use an expression cutoff (FPKM ≥ 2). However, a single dimension by using a FPKM cutoff alone is insufficient. Using a variance cutoff or developing statistical models for identifying highly variable genes had been used in prior publications, especially for small sample sizes and for scRNA-seq, to account for technical variability in library preparation and low read counts per cell (e.g. Law CW et al., *Genome Biology* 2014; Chen HH et al., *BMC Genomics* 2016).

We thank the reviewer for the clarification and we understand that the reviewer's concern also stems from having a standardized statistical justification across different studies. We had systematically compared the percentage of genes that overlap between the lists of genes with

high intra-individual variability and genes with high inter-individual variability. The thresholds that we used were selected based on the points of inflection for both intra-individual variability (SD=2) and inter-individual variability (SD=1.5), and we have now included the data in Supplementary Table 5 and plotted the results in the updated Supplementary Fig. 2.

Supplementary Figure 2. Varying intra-individual and inter-individual standard deviations (SD). (A) The ratios of overlap between the genes with high intra-individual SDs and high inter-individual SDs, with intra-individual SDs varying from 0.5 to 3. (B) The ratios of overlap between the genes with high inter-individual SDs and high intra-individual SDs, with inter-individual SDs varying from 0.5 to 3.

Reviewer 4 Major issues:

4. While the authors state that they have modified the text to clarify the differential expression much of the text presented as new, it is essential the same text from the original submission and the authors did not respond to our comment. - is there any overlap between the different sets (here we only see SetA compared with the 15q11-13 deletion). Are the logFC similar?

We thank the reviewer for raising this point. Actually there is a high correlation between the fold changes of 8 out of 9 genes in common between Set A of the 16p11.2 and 15q11-13 (Pearson's $r = 0.92$, $P = 0.0014$).

We moved all these analyses in the updated Supplementary Materials section, and expanded on the analyses:

“Comparison of differentially expressed genes from 16p11.2 deletion and 15q11-13 duplication cerebral organoids reveals 9 genes in common

We compared the differentially expressed genes with $FDR \leq 0.05$ between the 16p11.2 deletion SetA and 15q11-13 duplication results, and observed that there were 8 genes that were differentially expressed in the same direction for 16p11.2 deletions (SetA) and 15q11-13 duplications (*RPS14*, *PCDHGB6*, *TUBGCP5*, *CYFIP1*, *ELAVL2*, *SNHG5*, *NAP1L5* and *MYL6B*). There was a high correlation between the fold changes of these 8 genes between 16p11.2 deletions and 15q11-13 duplications (Pearson's $r=0.92$, $P=0.0014$).

Gene	Fold change in 16p11.2	Fold change in 15q11-13
RPS14	1.09	1.11
PCDHGB6	0.54	0.54
TUBGCP5	1.18	1.72
CYFIP1	1.05	1.35
ELAVL2	0.68	0.52

SNHG5	1.15	1.3
NAP1L5	0.75	0.53
MYL6B	0.97	0.92

Another gene (*HERC2*) was also differentially expressed for 16p11.2 deletions (SetA) and 15q11-13 duplications but in opposite directions. *HERC2* was over-expressed in 15q11-13 duplications cases compared to controls (fold change = 1.48), whereas *HERC2* was under-expressed in 16p11.2 deletion cases compared to controls (fold change = 0.9). Of the 9 genes that were differentially expressed, 3 of them (*TUBGCP5*, *CYFIP1* and *HERC2*) were found in the 15q11-13 locus. These results suggest that there are shared key genes that are perturbed by 16p11.2 deletions and 15q11-13 duplications.

There were 6 genes that were differentially expressed in the same direction for 16p11.2 deletions (SetP) and 15q11-13 duplications (*RPS14*, *PCDHGB6*, *ELAVL2*, *SNHG5*, *CTNNA2* and *NAP1L5*). There was a moderate correlation between the fold changes of these 6 genes between 16p11.2 deletions (SetP) and 15q11-13 duplications (Pearson's $r=0.73$, $P=0.064$).

Gene	Fold change in 16p11.2	Fold change in 15q11-13
RPS14	1.09	1.11
PCDHGB6	0.52	0.54
ELAVL2	0.72	0.52
SNHG5	1.15	1.3
CTNNA2	0.59	0.58
NAP1L5	0.69	0.53

HERC2 was also differentially expressed for 16p11.2 deletions (SetP) and 15q11-13 duplications but in opposite directions (fold change = 0.8 in 16p11.2 SetP and fold change = 1.48 in 15q11-13).

There were no significantly differentially expressed genes with $FDR \leq 0.05$ from the 16p11.2 deletion (SetD) analyses."

Reviewer 4 Major issues:

5. In this paper, organoids were grown for 46 days, while the cell types derived from single cell RNAseq were ascertained from 3 and 6 month-old organoids. The authors claim that their method is tolerant to these time differences, but in order to make this claim they need to show much more data, as this comprises very large differences in developmental stage and cell composition.

We agree with the reviewer that it is interesting to understand the transcriptomic differences during neurodevelopment and cell type composition between brain organoids with different ages. There are several scRNA-seq papers that have been published on brain organoids and fetal brains addressing these questions, and these questions are beyond the scope of our manuscript, so we did not specifically address them. We had also removed the comparisons between 3-month and 6-month organoids as they had been systematically compared in prior publications.

However, for the purpose of our manuscript, which is to identify cell type specific co-expressed driver genes in ASD, we demonstrate that our Orgo-Seq framework is robust in integrating scRNA-seq data from organoids with different ages (e.g. 3 and 6 month organoids), and bRNA-seq from 46-day old organoids in a few ways:

1. We integrated scRNA-seq data from the Tanaka study, which is an aggregated scRNA-seq dataset from 8 different brain organoid protocols and fetal brains across different developmental timepoints, and replicated our initial results when using scRNA-seq data from the Quadrato study (please see point 6 below).

2. We performed validation of one of the key findings to identify cell type specific co-expressed driver genes (*KCTD13*) using the mosaic organoid oFlowSeq framework.

3. We performed multiple analyses to compare our results with results reported by other studies (published and unpublished).

Reviewer 4 Major issues:

a. The authors do not show any evidence that the cell types that they are using in the analysis are even found at this early stage. For example - cluster c4 “cells with forebrain markers” - are these cells/markers present in the organoid data?

Yes, all the genes in each cell type cluster are expressed in bRNA-seq data from our 46-day old cerebral organoids. We had performed quality control to remove genes that are not expressed in the 46-day old cerebral organoids from all downstream analyses. For each of the cell type clusters, the genes that were found in the cell type cluster, and were expressed in our donor-derived cerebral organoids were reported in Supplementary Table 11. All the 10 cell type clusters from the Quadrato study were represented by multiple cell type specific genes (ranging from 47 to 266). All the 11 cell type clusters from the Tanaka study were represented by multiple cell type specific genes (ranging from 12 to 421).

Reviewer 4 Major issues:

b. One alternative hypothesis is that a difference in the cell specific markers could mean a developmental shift in the treated vs. control and not a cell enrichment shift. Can the authors show data for the cell specific markers across development to rule out this hypothesis or at least discuss this in interpreting their data?

Yes, we thank the reviewer for pointing this out, and we apologize that we did not articulate these hypotheses more clearly in our previous manuscript. We were also curious about these two alternative hypotheses, and our mosaic cerebral organoid experiments (oFlowSeq) served 2 main purposes:

- 1) to validate the critical cell type specific driver gene (*KCTD13*) based on our results from Orgo-Seq
- 2) to evaluate if there is a distortion in the proportion of Nestin+ cells due to *KCTD13* mutations.

If we did not identify differences in the proportions of Nestin+ cells for all deleterious mutations in *KCTD13*, then the results will suggest a developmental shift in transcriptomic differences within the cells and not differences in the proportions of cell types. However, we did observe a difference in the proportions of Nestin+ cells with *KCTD13* deleterious mutations, so we can rule out that the first hypothesis (intrinsic developmental shift). In addition, the distortion in cell type proportions is in concordance with prior literature on *KCTD13* deficiency (Kizner V *et al*, 2019), and might explain for the increased head sizes observed in patients with 16p11.2 deletions.

Our updated text reads:

“This validation approach provides an orthogonal approach to validate our observed results from Orgo-Seq, and also serves to test the two hypotheses generated from the Orgo-Seq results: if there is a distortion in the proportions of cell types between the cerebral organoids from cases

versus controls, or if there is an intrinsic developmental shift resulting in transcriptomic differences within the cells, but not a distortion in the proportions of cell types. If *KCTD13* mutations do not affect the proportions of a specific cell type population, we expect to observe that the mutations are not significantly enriched in cells that are positive or negative for that cell type marker (Fig. 4A, Supplementary Materials). However, if *KCTD13* mutations affect the proportions of a specific cell type population, we expect to observe that the mutations are significantly enriched in the cells that are positive for that cell type marker.”

Reviewer 4 Major issues:

6. The authors did not address the misclassification and while they claim that it was validated in another data set (from the same lab) there are many published single cell data sets and the authors should really validate their methods using many of these data sets.

We have now addressed the reviewer’s concerns about misclassification in 2 ways. Firstly, we have now re-done our Orgo-Seq analyses using a larger-scale aggregated dataset published by Tanaka *et al.*, and validated our initial results identifying *YPEL3*, *INO80E* and *KCTD13* as critical cell type specific driver genes for 16p11.2 deletions.

In addition, we revisited the reviewer’s comment about potential misclassification of the identities of cell type clusters from different scRNA-seq studies by different labs, and we developed an approach to enable consistent cell type identification across different studies, by using 2 neurodevelopmental reference maps to fine-map the cell type identities from different studies.

Our updated text is as follows:

Introduction

“We applied Orgo-Seq for two ASD-associated copy number variants (CNVs) in the 16p11.2 and 15q11-13 loci¹⁶⁻¹⁸, by integrating 3 sets of transcriptomics datasets: bRNA-seq data that we generated from donor-derived cerebral organoids, previously published scRNA-seq data from cerebral organoids and fetal brains^{1,13,19,20}, and previously published bRNA-seq data from human post-mortem brain samples in the BrainSpan Project²¹. Using an initial scRNA-seq dataset from 66,889 single cells¹, we initially observed that neuroepithelial cells are perturbed in donor-derived cerebral organoids from individuals with deletions in 16p11.2 compared to individuals without the deletions, and that 3 of the genes in the locus (*YPEL3*, *KCTD13* and *INO80E*) are likely to be candidate driver genes functioning in neuroepithelial cells. Using a larger scRNA-seq dataset comprising of 190,022 cells¹⁹ from brain organoids differentiated using 8 different protocols and fetal brains^{1,7-13,22}, and two neurodevelopmental maps constructed from scRNA-seq on brain organoids and fetal brains to fine-map the critical cell types^{13,20}, we replicated the critical cell type that was initially discovered, and were able to pinpoint the identity of the critical cell type more precisely during neurodevelopment to immature neurons and intermediate progenitor cells for the 16p11.2 locus. We also replicated our initial results that *YPEL3*, *KCTD13* and *INO80E* are cell type specific candidate driver genes.”

Results

“Fine-mapping of cell type identities using large-scale neurodevelopmental maps point to the role of immature neurons and intermediate progenitor cells for the 16p11.2 locus.

A recent study by Tanaka Y *et al.* had re-analyzed 190,022 cells from brain organoids differentiated using 8 different protocols and fetal brains^{1,7-13,22}, and identified 24 cell type clusters¹⁹. We systematically compared the percentage overlaps among genes across the 24 cell type clusters to identify 11 unique clusters (CC1-11; Supplementary Table 13). We calculated CellScores using the 11 cell type clusters for 16p11.2, and found that there was only

the cell type cluster comprising of cortical excitatory neurons (CC3) that had an $\text{FWER} \leq 0.05$ ($P(\text{CellScore}) = 3.6 \times 10^{-4}$, Fig. 3C). Interestingly, we did not observe any association for 16p11.2 with the neuroepithelial cell cluster (CC7) in the Tanaka study ($P(\text{CellScore}) = 0.11$).

To ensure consistencies in assigning cell type identities across different studies, and to fine-map the critical cell types more precisely during neurodevelopment, we used two neurodevelopmental maps that were reconstituted from scRNA-seq data on brain organoids and fetal brains. The first neurodevelopmental map by Velasco S *et al.* comprised of 12 cell types¹³, and we calculated the percentage overlap among genes from each cell type cluster reported by Quadrato G *et al.* and Tanaka Y *et al.*^{1,19} (Supplementary Table 14). We found that the neuroepithelial cell cluster (c9) from the Quadrato study overlapped most closely with immature projection neurons (mean overlap = 0.37%), and that the unknown cell type (c6) from the Quadrato study overlapped most closely with outer radial glia cells (mean overlap = 0.29%). The CC3 cluster from the Tanaka study overlapped most closely with immature projection neurons in the neurodevelopmental map (mean overlap = 1.14%), similar to the c9 cluster from the Quadrato study.

We used a second neurodevelopmental map by Eze UC *et al.* that comprised of 6 cell types²⁰, and calculated the percentage overlap among the genes from each cell type cluster reported by the Quadrato and Tanaka studies (Supplementary Table 14). We found that the c9 cluster from the Quadrato study overlapped most closely with the neuronal and intermediate progenitor cell clusters (mean overlaps = 8.1% and 5.4% respectively). Similarly, the CC3 cluster from the Tanaka study overlapped most closely with the neuronal and intermediate progenitor cell clusters (mean overlaps = 19.8% and 11.1% respectively). These results suggest that the critical cell types for the 16p11.2 locus are likely to be immature neurons and intermediate progenitor cells.

To evaluate the degree of independence among the genes in the c9 cluster from the Quadrato study and the CC3 cluster from the Tanaka study, we calculated the correlations between the mean overlaps across the two neurodevelopmental maps for both the c9 and CC3 clusters (Supplementary Table 15). We observed high correlations between both the c9 and CC3 clusters using both neurodevelopmental maps ($r=0.71$, $P=9.1 \times 10^{-3}$; $r=0.95$, $P=3.6 \times 10^{-3}$). However, there were stronger correlations between the CC3 and c5 clusters ($r=0.96$, $P=8.7 \times 10^{-7}$; $r=0.98$, $P=6.3 \times 10^{-4}$), even though the c5 cluster was not implicated as the critical cell type from the 16p11.2 donor-derived organoids. This indicates that there is likely to be independence among the genes implicating the c9 and CC3 clusters as critical cell types in the 16p11.2 locus.”

Reviewer 4 Major issues:

7. (Additional comment). For the odds ratio statistics comparing the isogenic with the patient-derived lines, is this statistic (page 9 of the response to reviewers) comparing the overlap in DEGs between the neural stem cells vs. the iNeurons? Or is this showing that the neural stem cell/patient derived cerebral organoid overlap is more significant than would be expected by chance? Make it clear what statistical comparisons are being done in the text and figure legends.

Yes, the odds ratios are comparing the overlaps in DEGs between the organoids and the NSCs versus iNs. We had also reported the 95% confidence intervals and Fisher's Exact Test P-values to evaluate if the overlaps were more significant than would be expected by chance (that is, the overlap in DEGs between donor-derived cerebral organoids with iNs is assumed to be the baseline null).

We have updated the text to clarify this:

These observations provide further evidence that the differentially expressed genes from the patient-derived cerebral organoids are significantly more similar to the differentially expressed genes from the isogenic NSCs than the isogenic iNs with the same 16p11.2 deletion or duplication than by chance.

We have also made a figure in the Supplementary Methods section to provide clarity to the results:

Reviewer 4 Minor issues not addressed (from critiques):

10. Using all samples, even those from the same individual, as independent samples in the linear model falsely inflates the number of DEG (see Germain and Testa 2017 Stem Cell Reports). Instead, a mixed linear model should be used to account for multiple samples coming from the same individual. - this was not addressed. The assumption of independence even if it was used to increase the number of permutations is wrong.

We have now performed a DEG analysis for 16p11.2 SetA using EdgeR with a linear mixed effects model and accounting for multiple replicates from the same individual. The histograms of the p-value and FDR distributions are shown below.

For each gene, we calculated a ratio, which gives us an estimate of the degree of inflation or deflation in the P-value distributions.

$$\text{Ratio} = -\log_{10}(\text{P-value from linear regression}) \text{ divided by } -\log_{10}(\text{P-value from EdgeR})$$

We plotted the $\log_{10}(\text{Ratio})$ histogram below:

As the reviewer correctly pointed out, there is an inflation and the median $\log_{10}(\text{Ratio})$ is 0.4156586 (indicated by the red line), and $10^{0.4156586} = 2.604106$.

Our *GeneScore* statistic is:

$$\text{GeneScore}(x) = \frac{1}{\log_{10}\lambda} \sum_{\text{all } y} \frac{-\log_{10} P_y \times r_{x,y}^2}{\text{Num}_y}$$

We had used the λ term in *GeneScore* to correct for any potential inflation in the p-value distributions. From the 16p11.2 SetA data, the calculated λ was 5.60565, which is higher than the median inflation calculated from comparing the two different DEG analyses (2.604106). As such, the test statistics that we had used were even more stringent than accounting for multiple replicates from the same individual alone.

We did not include these analyses mentioned above in our revised manuscript as we felt that these were tangential points that would cause confusion to the readers. However, we reiterate that we had used highly conservative statistics and analyses in our current Orgo-Seq manuscript as our initial manuscript is to evaluate if we can detect cell type specific co-expressed driver genes by integrating bRNA-seq and scRNA-seq data from multiple brain organoid studies, and we wanted to ensure that our results were robust and were likely to replicate (using the mosaic organoid oFlowSeq approach and when integrating additional scRNA-seq data).

Reviewer 4 Minor issues:

15. There are still more sections in the results that need to be moved to the discussion. We have revised the manuscript substantially and moved several sections into the supplementary materials. Also, we had removed several sections that rehashed the results from prior publications, and instead, we cited these prior publications instead.

Reviewers' comments:

Reviewer #3 (Remarks to the Author):

The revised manuscript, now called "Multi-transcriptomics data integration of cerebral organoids and post-mortem brains to identify cell type specific co-expressed driver genes in autism" has been considerably improved by the authors. The text and logic flow is improved to gain clarity. Inclusion of a new reference scRNAseq dataset, a combination of several recent datasets from multiple labs performed by Tanaka et al. is very valuable and firmly grounds Orgo-Seq within a variety of protocols and organoids. The authors have performed several additional computational analyses and controls bolstering their claims. I remain skeptical of the FACS-based verification, which still raises a lot of questions. Given that the paper's increased focus on the computational Orgo-Seq pipeline, the FACS-based assay seems dispensable and, in my mind, might be removed from the ms entirely. Given the extensive bulk RNA-Seq provided and more importantly the Orgo-Seq pipeline, which constitutes a valuable contribution in this very exciting, important and ongoing research area, I would recommend publication upon some editorial changes.

Major points:

1. The text still would benefit from further streamlining. As an example, there is a lengthy passage on the initial use of the Quadrato et al dataset as the scRNAseq reference for Orgo-Seq. Given that the authors have repeated this analysis using the much better Tanaka et al combined reference dataset, this should be the focus of the text.
2. Like before, the figures need some improvement. Font sizes are too small to read at times (ie figure 3, axis labelling, multiple supplementary figures).
3. As explained above, I would propose to remove the FACS data. However, if the authors insist to keep them, they need to explain their approach. a) As mentioned before, why are so few cells positive for the markers used? This is inconsistent with other lab's data on progenitor and neuron percentages in organoids. b) In the FACS plots (Supplementary figure 10), why is IgG2A used as a control, even though the antibodies are of a different antibody class? c) Why are the forward scatter plots so different? These are antibody independent and should be equal. d) All gates used are equal and presumably set on the incorrect IgG2A isotype control. The increased rates of positive cells for all the markers could be simply explained by the different antibody conditions used. In fact, the entire population (including negative cells, easily seen by the red mark in the heat map plots) is shifted to the right by about 0.5 logs on the x-axis. e) Were similar FACS settings used?

Reviewer #4 (Remarks to the Author):

The overall goals of this work are laudable and a lot has been done. There were serious critiques to the previous version and authors have done a good job of answering concerns especially regarding some of the more technical issues that concerned me and other reviewers. The paper is much better put together and clearer to follow. The major issue is whether the major claims vis a vis driver genes is convincing and whether Orgo-seq is itself an advanced and widely useful framework for integration of multi-transcriptomics data sets as claimed --- I am not entirely convinced, but the work is very good, and I also think that the authors have done a reasonable job responding (and this is a nice data set), so that this should be published. These issues will be decided by the readers and the community once the work is published, and as the field moves forward with larger, additional samples as suggested.

My minor comments are below:

The high reproducibility and demonstration that genes within the CNVs study change as would be expected from the dosage changes are important strengths of the work.

Other published work shows similar high correlation between organoids from controls and should be cited.

I do wonder whether pairwise correlations used to identify co-expression are nearly as robust as graph theoretical measures such as topological overlap (TO). It has been shown that TO removes spurious correlations and reduces noise, compared with standard pairwise correlation measures. In this case, TO would remove noisy genes or spuriously correlated genes -conversely this weakness with pairwise correlation may underlie the initial negative results with 15q. Perhaps applying these methods (see also recent Nature Genetics paper integrating single cell and bulk tissue- PMID: 34239132) using this more robust framework would yield stronger putative drivers at the 15q11-13 locus.

One issue that I had which is likely out of the scope of the paper is that there seems to be very low mean overlap between the clusters in the Quadrato and Tanaka studies with the Velasco neurodevelopmental maps (ranging from 0.37%=1.14%). It is also unclear from the methods how this mean overlap was calculated.

Another issue is the interpretation of the integration of the bulk RNAseq from the postmortem brains. While the method developed in the paper allows to integrate the organoid single cell and post mortem bulk RNAseq the two tissues have different cell types or at the very least different levels of maturation. To make this point even more apparent the top cell type cluster identified was the stem cell cluster which is massively depleted in the adult brain.

We thank both reviewers for their comments, and we summarize a few key revisions to our manuscript.

Orgo-Seq package and documentation

We have now re-implemented Orgo-Seq as an R package, provided the source codes, and wrote detailed documentation for running the package on our GitHub site (<https://gitlab.com/elimlab/orgo-seq>). As part of the re-implementation, we have also worked on optimizing the codes, parallelization and improving runtimes. Given the more efficient codes, we have now performed 1 million permutations for our 16p11.2 *CellScore* results, instead of the initial 100,000 permutations. We have also calculated *CellScores* for the 15q11-13 data using the Tanaka dataset, but similar to the Quadrato dataset, we did not observe any cell type clusters with $\text{FWER} \leq 0.1$ for the 15q11-13 data.

Figures

We have re-done most of the figures in the manuscript and the supplementary materials, and re-organized panels within the figures to improve readability.

oFlowSeq (FACS based system)

We have removed this section from our revised manuscript.

15q11-13 results

We re-ran our analyses for critical cell types using both the Quadrato and Tanaka datasets for the 15q11-13 locus (100,000 permutations). We did not observe any significant critical cell type using both datasets. To avoid confusion, we have revised our manuscript to focus only on the 16p11.2 driver gene results in the critical cell type identified, and removed the 15q11-13 putative driver gene results.

Our point-by-point responses (in blue text) to the reviewers' comments (in black text) is as follow. **References to updated text in the revised manuscript are highlighted in red text.**

Reviewer #3 (Remarks to the Author):

The revised manuscript, now called "Multi-transcriptomics data integration of cerebral organoids and post-mortem brains to identify cell type specific co-expressed driver genes in autism" has been considerably improved by the authors. The text and logic flow is improved to gain clarity. Inclusion of a new reference scRNAseq dataset, a combination of several recent datasets from multiple labs performed by Tanaka et al. is very valuable and firmly grounds Orgo-Seq within a variety of protocols and organoids. The authors have performed several additional computational analyses and controls bolstering their claims. I remain skeptical of the FACS-based verification, which still raises a lot of questions. Given that the paper's increased focus on the computational Orgo-Seq pipeline, the FACS-based assay seems dispensable and, in my mind, might be removed from the ms entirely. Given the extensive bulk RNA-Seq provided and more importantly the Orgo-Seq pipeline, which constitutes a valuable contribution in this very exciting, important and ongoing research area, I would recommend publication upon some editorial changes.

We thank the reviewer for his/her suggestions, and have now removed the FACS-based assay from the manuscript.

1. The text still would benefit from further streamlining. As an example, there is a lengthy passage on the initial use of the Quadrato et al dataset as the scRNAseq reference for Orgo-Seq. Given that the authors have repeated this analysis using the much better Tanaka et al combined reference dataset, this should be the focus of the text.

We thank the reviewer and have rewritten the text to state that we used the Quadrato and Tanaka datasets, and replicated our findings with both datasets.

2. Like before, the figures need some improvement. Font sizes are too small to read at times (ie figure 3, axis labelling, multiple supplementary figures).

We apologize and have re-done most of the figures to ensure that all axes are labeled, and re-organized the figures within multiple larger figures to provide better clarity.

3. As explained above, I would propose to remove the FACS data. However, if the authors insist to keep them, they need to explain their approach. a) As mentioned before, why are so few cells positive for the markers used? This is inconsistent with other lab's data on progenitor and neuron percentages in organoids. b) In the FACS plots (Supplementary figure 10), why is IgG2A used as a control, even though the antibodies are of a different antibody class? c) Why are the forward scatter plots so different? These are antibody independent and should be equal. d) All gates used are equal and presumably set on the incorrect IgG2A isotype control. The increased rates of positive cells for all the markers could be simply explained by the different antibody conditions used. In fact, the entire population (including negative cells, easily seen by the red mark in the heat map plots) is shifted to the right by about 0.5 logs on the x-axis. e) Were similar FACS settings used?

We have removed the FACS data from our current manuscript. We sincerely thank the reviewer for these helpful comments.

Reviewer #4 (Remarks to the Author):

The overall goals of this work are laudable and a lot has been done. There were serious critiques to the previous version and authors have done a good job of answering concerns especially regarding some of the more technical issues that concerned me and other reviewers. The paper is much better put together and clearer to follow. The major issue is whether the major claims vis a vis driver genes is convincing and whether Orgo-seq is itself an advanced and widely useful framework for integration of multi-transcriptomics data sets as claimed --- I am not entirely convinced, but the work is very good, and I also think that the authors have done a reasonable job responding (and this is a nice data set), so that this should be published. These issues will be decided by the readers and the community once the work is published, and as the field moves forward with larger, additional samples as suggested.

The high reproducibility and demonstration that genes within the CNVs study change as would be expected from the dosage changes are important strengths of the work.

We thank the reviewer for his/her encouragement.

Other published work shows similar high correlation between organoids from controls and should be cited.

We have included an additional reference comparing organoids from different donors.

I do wonder whether pairwise correlations used to identify co-expression are nearly as robust as graph theoretical measures such as topological overlap (TO). It has been shown that TO removes spurious correlations and reduces noise, compared with standard pairwise correlation measures. In this case, TO would remove noisy genes or spuriously correlated genes -conversely this weakness with pairwise correlation may underlie the initial negative results with 15q. Perhaps applying these methods (see also recent Nature Genetics paper integrating single cell and bulk tissue- PMID: 34239132) using this more robust framework would yield stronger putative drivers at the 15q11-13 locus.

We thank the reviewer for the interesting suggestion, and we are currently working on generating additional datasets to evaluate this question with different approaches. We have removed the section of putative cell type specific driver genes in the 15q11-13 locus, given that there was no critical cell type identified for the 15q11-13 locus.

One issue that I had which is likely out of the scope of the paper is that there seems to be very low mean overlap between the clusters in the Quadrato and Tanaka studies with the Velasco neurodevelopmental maps (ranging from 0.37%=1.14%). It is also unclear from the methods how this mean overlap was calculated.

We agree with the reviewer that additional research and investigation into datasets from scRNA-seq technologies will be needed. We have also updated our revised manuscript to clarify how we calculated the mean overlap.

Calculation of overlaps and correlations with neurodevelopmental maps

We used the cell type clusters from two large-scale scRNA-seq studies on brain organoids and fetal brains as neurodevelopmental maps^{13,20}, and calculated the percentage overlaps between the genes found in each cell type cluster from the Quadrato and Tanaka studies, and each cell type cluster from both neurodevelopmental maps (Supplementary Table 14). Using these percentage overlaps, we calculated the Pearson's correlations between each cell type cluster from both the Quadrato and Tanaka studies (Supplementary Table 15).

Another issue is the interpretation of the integration of the bulk RNAseq from the postmortem brains. While the method developed in the paper allows to integrate the organoid single cell

and post mortem bulk RNAseq the two tissues have different cell types or at the very least different levels of maturation. To make this point even more apparent the top cell type cluster identified was the stem cell cluster which is massively depleted in the adult brain.

We agree with the reviewer that additional research into the integration of bulk RNA-seq data from brain organoids and postmortem brains, will be needed. We have rewritten the section to emphasize that we did not detect any critical cell type through integration of bulk RNA-seq data from brain organoids and postmortem brains for the 15q11-13 locus.

Data integration of bRNA-seq data from post-mortem brain samples and scRNA-seq data from control cerebral organoids to identify critical cell types

A prior publication had performed RNA sequencing on post-mortem brain samples of the cortex that were obtained from 9 individuals with 15q11-13 duplications and 49 control individuals³⁹. We calculated *CellScores* for each of the 10 cell type clusters in the Quadrato study using the differential expression results from the post-mortem brain samples, and calculated a weighted average $P(\text{CellScore})$ using the results from the patient-derived cerebral organoids and post-mortem brain samples with 15q11-13 duplications (Supplementary Table 16). Similar to our results from the patient-derived cerebral organoids, there were no cell type clusters identified from the post-mortem brain samples that was significantly perturbed.